# The impact of locomotion on the brain evolution of squirrels and close relatives

Ornella C. Bertrand [1✉], Hans P. Püschel [1], Julia A. Schwab [1], Mary T. Silcox [2] & Stephen L. Brusatte[1]

How do brain size and proportions relate to ecology and evolutionary history? Here, we use virtual endocasts from 38 extinct and extant rodent species spanning 50+ million years of evolution to assess the impact of locomotion, body mass, and phylogeny on the size of the brain, olfactory bulbs, petrosal lobules, and neocortex. We find that body mass and phylogeny are highly correlated with relative brain and brain component size, and that locomotion strongly influences brain, petrosal lobule, and neocortical sizes. Notably, species living in trees have greater relative overall brain, petrosal lobule, and neocortical sizes compared to other locomotor categories, especially fossorial taxa. Across millions of years of Eocene-Recent environmental change, arboreality played a major role in the early evolution of squirrels and closely related aplodontiids, promoting the expansion of the neocortex and petrosal lobules. Fossoriality in aplodontiids had an opposing effect by reducing the need for large brains.

[1] School of GeoSciences, University of Edinburgh, Grant Institute, Edinburgh, Scotland, UK. [2] Department of Anthropology, University of Toronto Scarborough, Toronto, ON, Canada. ✉email: ornella.betrand@ed.ac.uk

What ecological and evolutionary factors affect brain size in mammals? Studies have assessed the impact of various ecologies and behaviours—such as diet, locomotion, habitat, and activity pattern—on relative brain size in various mammalian groups[1–12]. There is evidence that relatively larger brains are associated with diets involving more complex foraging strategies (vs. folivorous diets) (e.g.[2,10,11]), diurnal (vs. nocturnal) activity pattern[9] (but see[6]), life in complex environments like forests (vs. more open habitats) (e.g.[1,12]), and arboreal or climbing locomotion (vs. fossorial or digging) (e.g.[5,8,9]). Much of this work, however, was conducted several decades ago, did not take phylogenetic relationships into account, and used methods (such as filling the endocranial cavity with beads to measure volume) that could not tease apart the relative sizes of important brain regions, like the olfactory bulbs involved in smell, the neocortex that controls sensory processing and integration, and the petrosal lobule, involved in gaze stabilization and motor control[13–15] (Fig. 1).

More recent studies have allowed for a more nuanced parsing out of functional regions by using fresh brains or endocasts, often rendered from computed tomography (CT) scans. Some studies find correlations between certain brain regions and behaviour—such as larger olfactory bulbs in frugivorous and nocturnal mammals compared to folivores, diurnal taxa, and aquatic carnivores (e.g.[16–19]) —whereas others reveal little correlation between ecology and the relative size of the neocortex[12,20] and petrosal lobule[21]. Many of these studies, however, focused on extant mammals only, without taking extinct species into account. In general, few studies combine modern and fossil species, and test how brain size and shape and the relative proportions of brain regions changed over time, across phylogeny, and in relation to ecology or behaviour[22–27].

Rodents are an ideal group for such a study. They are ubiquitous today with more than 2500 species, making up ca. 40% of mammalian diversity[28]. They exhibit a vast range of behaviours, habitats, and locomotor abilities, which they developed over a long evolutionary history stretching back to the late Paleocene, ~57 million years ago[29,30]. Squirrels (Sciuridae), the third most diverse extant family of rodents, display an especially wide array of lifestyles, including terrestrial, scansorial, arboreal, and gliding forms[31]. On the other hand, the closest living relative of squirrels, the mountain beaver, is a small fossorial rodent endemic to western North America[32,33]. It is the only remaining member of Aplodontiidae, a once diverse clade with nearly 100 species that included the now-extinct fossorial horned gophers[33,34]. The striking differences between living sciurids and aplodontiids raise the question of how this modern rodent assemblage emerged.

The likely common ancestors of squirrels and the mountain beaver are known as ischyromyids[35]. They diversified during the early Eocene greenhouse climate (ca. 56–48 million years ago) and were likely fossorial to scansorial[36,37], but then declined and went extinct by the end of the Oligocene (23 million years ago)[38]. The larger squirrel + aplodontiid group (Sciuroidea) first appeared in the fossil record during the late Eocene, the time of a major cooling event[31,33,39]. During the Oligocene, for about 11 million years, ground and tree squirrels and aplodontiids with similar lifestyles and habitats radiated across woodland environments of the Holarctic region[40]. Then, in the drier and more open environment of the Miocene (23–5.3 million years ago), aplodontiids shifted to become predominantly specialized diggers, while squirrels continued to occupy the same Oligocene niches, and gliding evolved in some species[41–43]. By the end of the Miocene, aplodontiid diversity had dramatically declined, perhaps related to the spread of grasslands, a more favourable habitat for ground squirrels[43,44]. Aplodontiids are then absent from the fossil record until the late Pleistocene appearance of *Aplodontia rufa*, suggesting that this group was becoming less diverse, while squirrels showed the opposite pattern[43,45].

Previous work suggests that one particular aspect of ecology—locomotor behaviour—might be reflected in the brains of rodents. A relationship between relatively large brains and arboreality has been established for rodents generally[9,11] and squirrels specifically[8] (but see[7]), and there is evidence that the sizes of the neocortex, olfactory bulbs, and petrosal lobules vary widely among these animals and might be tied in some cases to locomotor differences[26,46]. However, this work did not account for phylogeny and the effect of body mass when assessing the correlation between locomotion and brain size and did not assess the relative sizes of different brain regions together in the same analysis.

Here we use a CT scan dataset of extinct and extant squirrels, aplodontiids, and close relatives, and assess changes in the brain over time, across phylogeny, and associated with locomotor behaviour. Our data track the evolution of squirrels and close relatives over ~53 million years of evolution, a time of tremendous global changes including warming and cooling events and the spread of grasslands and glaciers and illuminates how the brains and sensory systems of today's species developed. We use our data to address the following key questions: (1) Do locomotion, body mass, and phylogeny correlate with overall brain

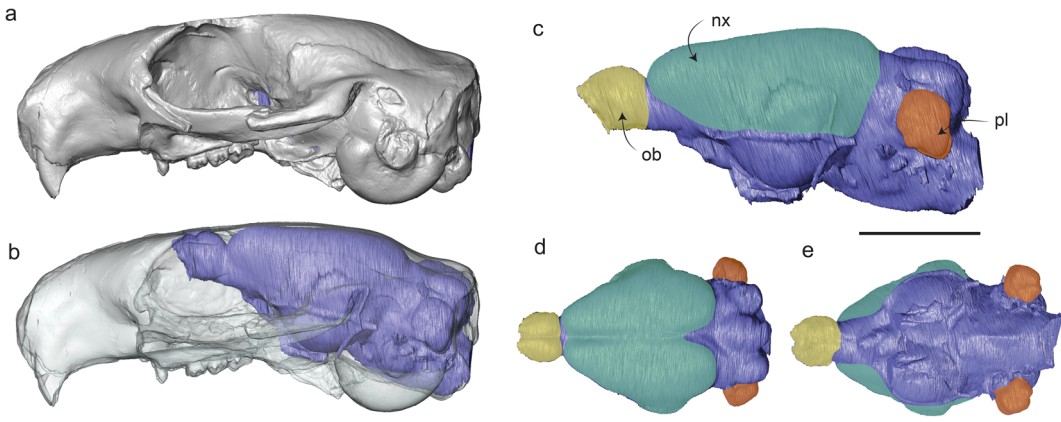

**Fig. 1 Virtual endocast of the early Oligocene squirrel *Cedromus wilsoni* (USNM 256584) based on computed tomography (CT) data. a** lateral view inside a solid cranium; **b** lateral view inside a translucent cranium; **c** lateral view with the different regions estimated in the paper highlighted; **d** dorsal view; **e** ventral view. nx neocortex, ob olfactory bulb, pl petrosal lobule. Scale bars equal 10 mm.

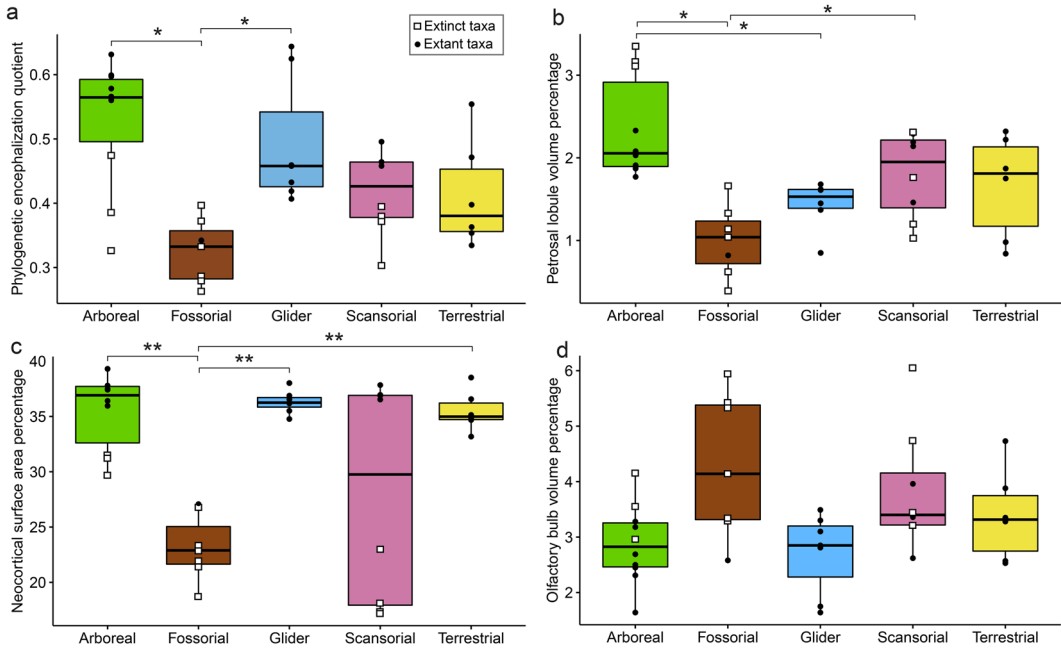

**Fig. 2 Phylogenetic Encephalization Quotient and brain component percentage comparisons among locomotor groups for Sciuroidea.** Boxplots of the (**a**) Phylogenetic Encephalization Quotient (PEQ); **b** Neocortical surface area percentage; **c** petrosal lobule volume percentage; **d** olfactory bulb volume percentage for our sample of rodents categorized by locomotor mode. * are for categories that are significantly different based on the Asymptotic K-Sample Fisher-Pitman Permutation Test (Table 1). $n = 38$ for **a**, **b**, and **d** and $n = 37$ for **c**.

size and the size of brain regions? (2) Do arboreal species have larger brains than others, particularly fossorial species, after body size and phylogeny are taken into account? (3) What sequence of changes occurs, over time and across phylogeny, as a rodent becomes arboreal or fossorial?

Our analyses show that locomotion in association with body mass is significantly related to endocranial and petrosal lobule volumes and that locomotion alone has a significant effect on the relative size of the petrosal lobules and neocortex in extinct and extant rodents. We show that specific regions of the brain (i.e., neocortex and the petrosal lobules) underwent profound modifications potentially intertwined with changes in locomotor behaviour throughout the Cenozoic, across ca. 53 million years of evolution during a time of dynamic global change. More specifically, our study allows us to recognize that separate radiations of burrowers and tree-dwellers independently evolved the brain features characteristic of these behaviours.

## Results

Our Fisher-Pitman Permutation Test, pairwise comparisons, and boxplots show that arboreal species (Fig. 1) and gliders have significantly higher PEQ compared to fossorial species (Fig. 2a; Table 1). The petrosal lobule volume percentage is significantly higher in arboreal compared to fossorial and gliding taxa (Fig. 2b). Additionally, scansorial taxa have significantly bigger petrosal lobules compared to fossorial species (Fig. 2b). Arboreal, gliding, and terrestrial species have significantly bigger neocortices compared to fossorial taxa and do not overlap in range with the latter (Fig. 2c). Scansorial taxa overlap with all other locomotor categories in neocortical surface area. More specifically, extant scansorial members show similar values to other extant squirrels, whereas extinct scansorial taxa are in the range of fossorial species for the neocortical surface area percentage (Fig. 2c). Extinct arboreal taxa are closer to the range of extant groups for the neocortex compared to extinct scansorial and fossorial species (Fig. 2c). All locomotor groups overlap when the olfactory bulb volume percentage is assessed (Fig. 2d).

The six PGLS regressions with their selected associated model and formula are shown in Table 2. In all regressions, Predictor 1 (Body mass in Set 1 and Endocranial volume in Set 2) is significantly correlated with all response variables (i.e., olfactory bulbs, petrosal lobules, and neocortex) ($p = 0.000$; Table 2). For set (1) in which body mass and locomotion are considered predictors, the PGLS regression results show that the interaction between body mass and locomotion has a significant effect on endocranial volume ($p = 0.024$), whereas locomotion alone does not ($p = 0.121$; Table 2). The petrosal lobule volume is significantly explained by the interaction between locomotion and body mass ($p = 0.027$) but not locomotion alone ($p = 0.056$; Table 2). For set (2) where endocranial size and locomotion are the predictors, we find that locomotion alone can significantly predict the size of the petrosal lobules and neocortex ($p = 0.009$ and 0.001 respectively; Table 2). In regressions of both set (1) and (2), we failed to find a significant impact of locomotion on the olfactory bulb volume ($p > 0.05$; Table 2). In terms of the strength of the phylogenetic signal for each regression, it is the highest for the endocranial volume and neocortical surface area regressions ($ƛ > 0.72$), while the remaining regressions (i.e., olfactory bulbs and petrosal lobules) have $ƛ$ values below 0.7 suggesting a weaker phylogenetic signal (Table 2).

Graphically, similar results to the permutation test are visible for the six PGLS regressions in Fig. 3. We observe that arboreal and gliding rodents have greater endocranial volumes compared to other locomotor modes (Fig. 3a). Fossorial and scansorial rodents have relatively smaller neocortices compared to other lifestyles (Fig. 3b). Arboreal species have relatively larger petrosal lobules compared to fossorial and gliding taxa. Additionally, fossorial rodents have relatively smaller petrosal lobules compared to scansorial species (Fig. 3). When the petrosal lobule volume is considered against body mass rather than endocranial volume, we observe that gliding taxa have a higher slope, leading to large gliders converging toward arboreal taxa while the opposite is true for small-sized gliders (Fig. 3c). The petrosal lobules also appear absolutely larger in arboreal and scansorial

**Table 1 . Locomotion permutation test and pairwise comparisons performed on the Phylogenetic Quotient (PEQ) and brain regions in Sciuroidea.**

| | PEQ | | | Petrosal lobules | | |
|---|---|---|---|---|---|---|
| Permutation test | $\chi^2 = 17.166$, df = 4, p < 0.002 | | | $\chi^2 = 17.39$, df = 4, p < 0.002 | | |
| Pairwise comparisons | Stat | p.value | p.adjust | Stat | p.value | p.adjust |
| Arboreal - Fossorial | 3.143 | 0.002 | **0.017** | 3.163 | 0.002 | **0.016** |
| Arboreal - Glider | 0.746 | 0.456 | 0.506 | 2.603 | 0.009 | **0.042** |
| Arboreal - Scansorial | 2.319 | 0.020 | 0.051 | 1.873 | 0.061 | 0.100 |
| Arboreal - Terrestrial | 2.065 | 0.039 | 0.078 | 1.935 | 0.053 | 0.100 |
| Fossorial - Glider | -2.721 | 0.007 | **0.033** | -1.812 | 0.070 | 0.100 |
| Fossorial - Scansorial | -2.414 | 0.016 | 0.051 | -2.498 | 0.012 | **0.042** |
| Fossorial - Terrestrial | -1.987 | 0.047 | 0.078 | -1.953 | 0.051 | 0.100 |
| Glider - Scansorial | 1.650 | 0.099 | 0.141 | -1.474 | 0.141 | 0.176 |
| Glider - Terrestrial | 1.459 | 0.145 | 0.181 | -0.836 | 0.403 | 0.448 |
| Scansorial - Terrestrial | 0.106 | 0.916 | 0.916 | 0.457 | 0.647 | 0.647 |
| | Neocortex | | | Olfactory bulbs | | |
| Permutation test | $\chi^2 = 19.879$, df = 4, p < 0.001 | | | $\chi^2 = 11.424$, df = 4, p < 0.022 | | |
| Pairwise comparisons | Stat | p.value | p.adjust | Stat | p.value | p.adjust |
| Arboreal - Fossorial | 3.569 | 0.000 | **0.003** | -2.416 | 0.016 | 0.112 |
| Arboreal - Glider | -0.519 | 0.604 | 0.671 | 0.479 | 0.632 | 0.632 |
| Arboreal - Scansorial | 2.045 | 0.041 | 0.101 | -2.014 | 0.044 | 0.112 |
| Arboreal - Terrestrial | 0.081 | 0.935 | 0.935 | -1.296 | 0.195 | 0.278 |
| Fossorial - Glider | -3.441 | 0.001 | **0.003** | 2.284 | 0.022 | 0.112 |
| Fossorial - Scansorial | -1.217 | 0.223 | 0.319 | 0.773 | 0.439 | 0.488 |
| Fossorial - Terrestrial | -3.242 | 0.001 | **0.004** | 1.403 | 0.161 | 0.268 |
| Glider - Scansorial | 1.956 | 0.050 | 0.101 | -2.007 | 0.045 | 0.112 |
| Glider - Terrestrial | 1.014 | 0.311 | 0.389 | -1.490 | 0.136 | 0.268 |
| Scansorial - Terrestrial | -1.678 | 0.093 | 0.156 | 0.822 | 0.411 | 0.488 |

**Abbreviation:** df, degrees of freedom; Stat, statistics.
Significant p-values are highlighted in bold.

rodents compared to fossorial taxa (Fig. 3c). There is no clear separation between the different locomotor modes when the olfactory bulb volume is plotted against endocranial volume or body mass (Fig. 3e, f).

The results from the non-phylogenetic ANOVA show that for set (1), locomotion alone has a significant impact on endocranial volume ($p = 0.001$; Supplementary Table 1). This result is different from the PGLS analysis in which only the interaction between locomotion and body mass had a significant impact on brain size. We found a similar pattern for the petrosal lobule volume in which locomotion alone can explain petrosal lobule size variation when phylogeny is not considered ($p = 0.001$; Supplementary Table 1). For set (2), when endocranial size and locomotion were the predictors of the petrosal lobule volume and neocortical surface area, we found that locomotion alone could predict the size of these two variables ($p < 0.005$; Supplementary Table 1). These results were also recovered in the PGLS analysis (Table 2). For both sets of regressions of the non-phylogenetic ANOVA, locomotion alone can predict the relative size of the olfactory bulbs ($p = 0.019$; Supplementary Table 1), whereas it was not the case when phylogeny was accounted for (Table 2).

The non-phylogenetically corrected post-hoc pairwise tests reveal similar patterns to the permutation test and PGLS regressions in which locomotor categories can be compared (Table 3). Arboreal and gliding taxa are significantly different from fossorial species in relative brain size as in the permutation test (=higher PEQ than fossorial rodents), and they are also different from scansorial and terrestrial groups as in the regression (=relatively larger endocranial volume compared to

scansorial and terrestrial taxa) (Fig. 3a). Fossorial taxa are significantly different from arboreal and scansorial categories for the relative size of the petrosal lobules (Table 3) as in the permutation test and in the PGLS regressions (=relatively smaller petrosal lobules compared to arboreal and scansorial taxa) (Table 1; Figs. 2 and 3). Gliders are also significantly different from arboreal taxa in terms of the relative petrosal lobule size as found in the permutation test and PGLS regressions (=relatively smaller petrosal lobules than arboreal species) (Tables 1 and 3; Figs. 2 and 3). Gliders are not significantly different from arboreal rodents when the petrosal lobule volume is plotted against body mass, as large gliders plot very closely to arboreal taxa in the PGLS analysis (large gliders have absolutely larger petrosal lobules compared to small-sized gliders) (Tables 1 and 3; Figs. 2 and 3). Fossorial and scansorial taxa significantly differ from arboreal, gliding, and terrestrial species in term of relative neocortical size, which is similar to the permutation test results and PGLS regressions (=relatively smaller neocortex than arboreal, gliding, and terrestrial taxa) (Tables 1 and 3; Figs. 2 and 3). No significant results or visual trends were found for the relative size of the olfactory bulbs in the permutation tests and PGLS regressions (Tables 1 and 3; Figs. 2 and 3). This contrasts with the post-hoc tests in which both arboreal and gliding species are significantly different from fossorial taxa (=relatively smaller olfactory bulbs compared to fossorial species) (Table 3; Fig. 3). Additionally, gliders and scansorial taxa differ significantly for the same regression (=relatively smaller olfactory bulbs compared to scansorial taxa) (Table 3; Fig. 3). Post-hoc tests were not performed on the olfactory bulb volume against body mass because we found no

**Table 2 Results from the PGLS regression analyses including the effect of body mass and locomotion for Sciuroidea.**

| Regression | Model - PGLS | AIC | Intercept | slope (Pred. 1) | slope (Pred. 2) | slope (Pred.1*Pred. 2) | P-value (Pred. 1) | P-value (Pred. 2) | P-value (Pred. 1*Pred. 2) | Lambda | RSE | Df |
|---|---|---|---|---|---|---|---|---|---|---|---|---|
| 1.EB | Lambda | −79.080 | −1.075 | 0.691 | 0.077 | −0.042 | **0.000** | 0.121 | **0.024** | 0.722 | 0.103 | 38, 34 |
| 1.PB | Lambda | −20.372 | −1.986 | 0.705 | 0.446 | −0.091 | **0.000** | 0.056 | **0.027** | 0.535 | 0.189 | 37, 33 |
| 2.PE | Lambda | −30.028 | −1.115 | 0.825 | −0.046 | — | **0.000** | **0.009** | — | 0.544 | 0.171 | 37, 34 |
| 2.NE | Brownian | −120.954 | −0.503 | 0.976 | −0.014 | — | **0.000** | **0.001** | — | 1.008 | 0.130 | 38, 35 |
| 1.OB | Ornstein-Uhlenbeck | −47.930 | −0.625 | 0.511 | −0.014 | — | **0.000** | 0.332 | — | 0.293 | 0.122 | 38, 35 |
| 2.OE | Lambda | −50.479 | −0.927 | 0.862 | 0.017 | — | **0.000** | 0.182 | — | 0.690 | 0.149 | 38, 35 |

1.EB, Endocranial volume – Body mass + Locomotion * Body mass + Locomotion; 1.PB, Petrosal lobule volume – Body mass + Locomotion * Body mass + Locomotion; 2.NE, Neocortex surface area – Endocranial surface area + Locomotion; 1.OB, Olfactory bulb volume – Body mass + Locomotion; 2.OE, Olfactory bulb volume – Endocranial volume + Locomotion; AIC, Akaike Information Criterion; Df, degrees of freedom; RSE, Residual standard error. Significant p-values are highlighted in bold.

significant effect of locomotion on this regression in the non-phylogenetic corrected ANOVA (Supplementary Table 1).

Our ancestral state reconstructions show substantial changes in PEQ, residuals, brain size, body mass, and in the proportion of the olfactory bulbs, petrosal lobules, and neocortex through time. Based on our reconstruction, the PEQ decreased in *Pseudotomus*, associated with a massive increase in both brain size and body mass within Ischyromyidae (Fig. 4; Supplementary Table 2). In contrast, the PEQ, brain size, and body mass all appear to have decreased along the branches leading to the group *Ischyromys* + Sciuroidea. We observe a shift occurring in Sciuroidea, with a PEQ increase associated with a stronger reduction in body mass and an increase in brain size compared to previous nodes. In comparison, the PEQ increase observed in squirrels was mainly associated with an increase in brain size (Fig. 4; Supplementary Figs. 1 and 2a; Supplementary Table 2). In both Sciurinae tribes (Sciurini and Pteromyini), PEQ continued to increase. Our ancestral state reconstruction shows that this PEQ elevation was related to a brain size and body mass increase in Sciurini, while the opposite was true for Pteromyini. At the base of Callosciurinae and Xerinae, no major change in PEQ was observed; however, the PEQ appears to have decreased in Marmotini (Xerinae) due to an increase in both brain size and body mass (Fig. 4; Supplementary Figs. 1 and 2a; Supplementary Table 2). Within Aplodontiidae, our results show that the PEQ, brain size, and body mass increased from early to more derived aplodontiids (i.e., Homalodontia; Fig. 4).

The neocortex accounted for 18.3% of the total endocranial surface area in the reconstructed common ancestor of our sample (Fig. 5a). We observe two independent neocortical increases: within Ischyromyidae in *Pseudotomus* (21%) and at the node leading to *Ischyromys* and Sciuroidea (26%). Our results show that at the Sciuroidea node, the neocortex expanded to 27.9% and then again independently on the lineages leading to Sciuridae (30.3%) and Aplodontiidae (29.1%). Based on our analyses, the neocortex continued to expand gradually within Sciuridae to about 36% for all nodes leading to extant lineages (Fig. 5a). We observe a neocortical decrease within ground squirrels in Marmotini and within Aplodontiidae, in Homalodontia (Fig. 5a).

The common ancestor of our sample is inferred to have had a petrosal lobule volume ratio of 1.25% (Fig. 5b). A size decrease in the petrosal lobules was observed within ischyromyids in *Pseudotomus* (1.03%). Based on our analyses, the proportion of the petrosal lobules gradually increased along the branches leading to Sciuridae and Aplodontiidae, reaching about 3% (Fig. 5b). Our results show that the petrosal lobules decreased in squirrels, with the most pronounced reduction displayed by flying squirrels with a ratio of 1.46% (Fig. 5b; Supplementary Table 2). A decrease appears to also have occurred in ground squirrels (Marmotini) and in Aplodontiidae, at the base of Homalodontia (Fig. 5b).

The olfactory bulbs of the common ancestor of our sample represented 5.8% of endocranial volume (Supplementary Fig. 2b). A strong decrease in the olfactory bulb volume was observed at the node leading to Reithroparamyinae + *Ischyromys* + Sciuroidea (3.5%). Then, the proportion of the olfactory bulbs remained relatively stable on the branches leading to squirrels and aplodontiids, and after they split (Supplementary Fig. 2b; Supplementary Table 2).

The locomotor behavioural ancestral state reconstruction showed that the common ancestor of rodents was most likely scansorial (Fig. 6). Based on our results, among Ischyromyidae, *Pseudotomus* and *Ischyromys* independently became more fossorial compared to other taxa. The common ancestor of *Ischyromys* + Sciuroidea appears to have been on its way to becoming arboreal (~75% probability), while the common ancestor of Sciuroidea was arboreal (Fig. 6). Our analysis shows that

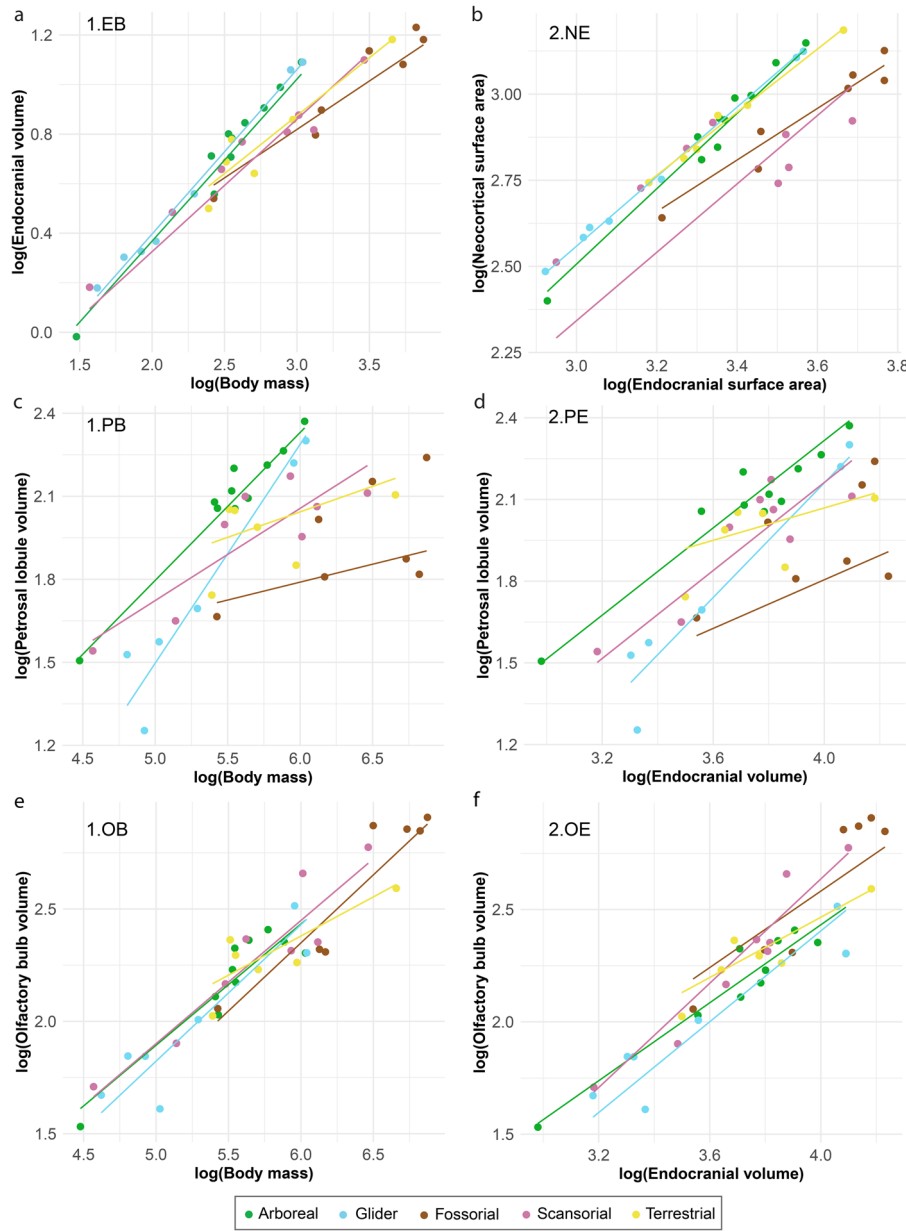

**Fig. 3 Relationships between the brain, brain regions, and body size organized by locomotor behaviour in Sciuroidea.** Phylogenetic corrected PGLS regressions of **a** 1.EB, Endocranial volume ~ Body mass + Locomotion * Body mass; **b** 2.NE, Neocortex surface area ~ Endocranial surface area + Locomotion; **c** 1.PB, Petrosal lobule volume ~ Body mass + Locomotion * Body mass; **d** 2.PE, Petrosal lobule volume ~ Endocranial volume + Locomotion; **e** 1.OB, Olfactory bulb volume ~ Body mass + Locomotion; **f** 2.OE, Olfactory bulb volume ~ Endocranial volume + Locomotion. Data used for all regression are in Logarithm 10. $n = 38$ for **a**, **b**, and **e**, **f**. and $n = 37$ for **c**, **d**.

Aplodontiidae and Sciuridae were ancestrally arboreal with fossoriality appearing later in aplodontiids (Fig. 6). Based on this reconstruction, the earliest Sciurini were arboreal and gliding evolved at the base of Pteromyini. Based on our sample, we reconstruct Callosciurinae and Xerinae as ancestrally terrestrial and the other lifestyles (i.e., arboreality and scansorial) secondarily evolved in these groups (Fig. 6).

## Discussion
Previous studies have found that larger relative brain size was more likely to be displayed by arboreal/climbing rodents (e.g.[8,9,11]) and species living in more complex environments such as forested habitats (e.g.[1,6,8]). Considering fossorial species, past studies have found that they displayed a relatively smaller brain

compared to other locomotor modes (e.g.[3,6,9,11]). It was also recognized that derived aplodontiids had relatively smaller brains compared to extant squirrels, which was suggested to be associated with their specialized digging lifestyle[26]. Our findings show that tree-dwellers living in closed habitats (i.e., arboreal and gliding taxa) have bigger relative brain size and higher PEQ compared to fossorial taxa, and to some degree compared to scansorial and terrestrial species. These results suggest that living in a more cluttered and complex environment such as the trees might require more neurological power and therefore may favour relatively larger brains, as previously hypothesized.

The neocortex controls an array of behaviours and its expansion may have allowed for a diverse range of advantages such as enhanced vision, memorization, sensory integration, and motor functions, resulting in higher level of data processing that could

**Table 3 . Non-phylogenetically corrected post-hoc pairwise comparisons P-values for the different locomotor modes in Sciuroidea.**

| Regression 1.EB | Arboreal | Glider | Scansorial | Terrestrial | Summary |
|---|---|---|---|---|---|
| Glider | 0.523 | | | | Arboreal & Glider: Relatively larger brain |
| Scansorial | **0.001** | **0.003** | | | |
| Terrestrial | **0.009** | **0.020** | 0.803 | | |
| Fossorial | **0.001** | **0.011** | 0.806 | 0.980 | |
| **Regression 1.PB** | | | | | |
| Glider | 0.239 | | | | Large gliders: absolutely larger PL compared to small gliders |
| Scansorial | 0.166 | **0.004** | | | |
| Terrestrial | **0.026** | **0.002** | 0.260 | | |
| Fossorial | 0.064 | **0.001** | 0.518 | 0.606 | |
| **Regression 2.PE** | | | | | |
| Glider | **0.008** | | | | Fossorial & Glider: Relatively smaller PL. Arboreal & Scansorial: Relatively larger PL |
| Scansorial | 0.131 | 0.224 | | | |
| Terrestrial | 0.086 | 0.420 | 0.751 | | |
| Fossorial | **0.001** | 0.401 | **0.027** | 0.068 | |
| **Regression 2.NE** | | | | | |
| Glider | 0.749 | | | | Fossorial & Scansorial: Relatively smaller neocortex |
| Scansorial | **0.005** | **0.019** | | | |
| Terrestrial | 0.967 | 0.766 | **0.010** | | |
| Fossorial | **0.003** | **0.022** | 0.635 | **0.003** | |
| **Regression 2.OE** | | | | | |
| Glider | 0.595 | | | | Arboreal & Glider: Relatively smaller OB |
| Scansorial | 0.063 | **0.020** | | | |
| Terrestrial | 0.280 | 0.160 | 0.542 | | |
| Fossorial | **0.007** | **0.010** | 0.445 | 0.179 | |

Abbreviations: 1.EB, Endocranial volume - Body mass + Locomotion * Body mass; 1.PB, Petrosal lobule volume - Body mass + Locomotion * Body mass; 2.PE, Petrosal lobule volume - Endocranial volume + Locomotion; 2.NE, Neocortex surface area - Endocranial surface area + Locomotion; 2.OE, Olfactory bulb volume ~ Endocranial volume + Locomotion; OB, olfactory bulbs; PL, petrosal lobules.
Significant p-values are highlighted in bold

be useful when inhabiting a highly complex environment like the forest canopy. However, previous studies have failed to find a correlation between locomotor behaviour and neocortical size. Instead, they found that neocortical size was correlated with social group size in primates and with sociality in euuungulates[12,20]. The extant mountain beaver has a lower neo-cortical ratio compared to its arboreal ancestor *Prosciurus*, sug-gesting that fossoriality may have a stronger influence on neocortical size than other locomotor behaviours do. Indeed, several studies have indicated that fossoriality may have a pro-found impact on the nervous system. A decrease in relative brain size was observed in fossorial golden moles compared to their ancestors[23], and a smaller neocortex and reduced caudal region of the neocortex devoted to vision (visual cortex) were noted in fossorial species such as golden moles and shrews[23,47–49]. Because the amount of light is limited underground, vision might not be as crucial. Regions that are less beneficial for the survival of the animal might not be selected because large brains are expensive to maintain[50]. Therefore, the visual cortex may have reduced in size in response to a specialization to fossoriality.

Our results show that all locomotor categories except fossorial overlap for neocortical size, which suggests that the size of the neocortex might not be enough to differentiate arboreal, gliding, scansorial, and terrestrial categories. Previous research has shown that tree squirrels had a more expanded visual cortex compared to terrestrial squirrels, while the latter exhibited more developed somatosensory and motor areas[51]. Arboreal squirrels require enhanced vision to navigate through the canopy, which would imply a more developed visual cortex compared to species spending more time on the ground. In Marmotini, the inferred neocortical decrease is very small (node 18 to 23; Supplementary Table 2) compared to the one observed in the fossorial mountain beaver. In fact, the maintenance of fairly large neocortices in spite of their terrestrial locomotor behaviour might be related to the high level of social behaviour displayed by ground squirrels[52,53]. Therefore, some areas of the neocortex related to communication in the pre-frontal cortex might be more developed in social ground-dwelling species compared to solitary tree-dwelling squirrels. This result suggests that not overall neocortical size, but rather the size of specific regions of the neocortex, might be related to locomotor behaviour and sociality. These neocortical differences might be impossible to identify on an endocast. However, if the relationship between an expanded visual cortex and arboreality can be supported for a large sample of extant squirrels, a stronger case could be made for extinct taxa that displayed an arboreal lifestyle.

The petrosal lobules are a specific region of the cerebellum that contributes to the stabilization of eye positions and movements as well as head coordination[15]. This function might be crucial for arboreal rodents, living in a multifaceted 3-dimensional space[54]. In general, fossorial rodents may require less control over their gait because they spend a considerable amount of time under-ground and may be less active compared to arboreal animals[26]. Our results show that fossorial rodents have on average absolutely and relatively smaller petrosal lobules compared to any other locomotor mode. Additionally, our ancestral state reconstruction illustrates that these changes occurred independently in two dif-ferent rodent lineages, Ischyromyidae and Aplodontiidae, as they

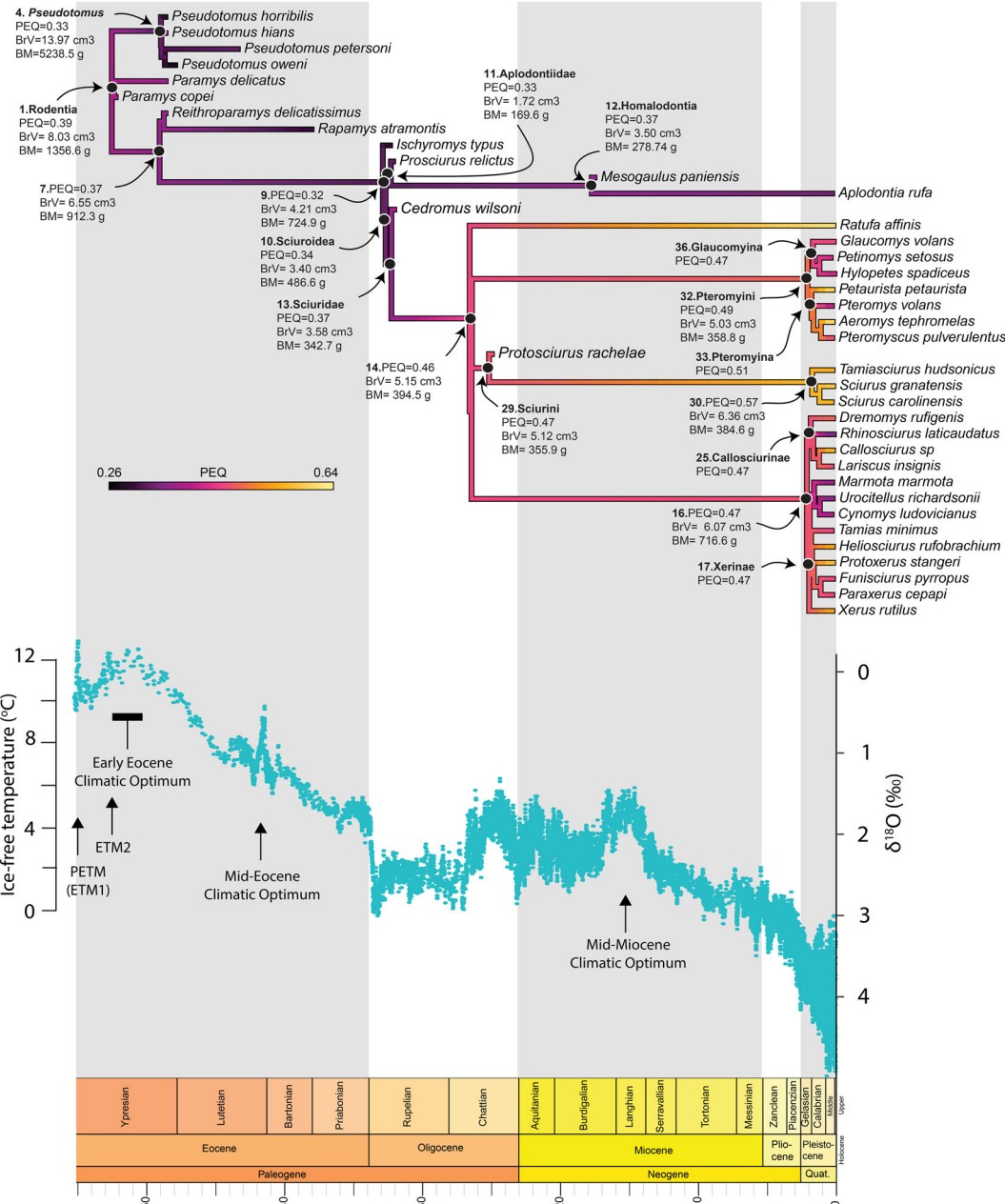

**Fig. 4 Ancestral state reconstruction of the Phylogenetic Encephalization Quotient (PEQ) with a geological time scale.** The temperature and oxygen level (δ18O) plot during the Cenozoic is modified from Zachos et al.[40].

became fossorial. These results conflict with a recent study that found no relationship between petrosal lobules and ecology in mammals[21]. The differing results may relate to the difference in sampling strategy, with the current work including a larger range of taxa that are closely related but that show a diversity of behaviours.

Knowing the role of the petrosal lobules in maintaining the position of the eyes when the head rotates (vestibulo-ocular reflex) and in the tracking of moving object (smooth pursuit)[55,56] leads us to hypothesize that these functions may be crucial when living in the 3D complex environment of the trees. Arboreal and scansorial taxa have relatively, and absolutely, larger petrosal lobules compared to fossorial taxa. The fact that some extant scansorial and terrestrial species overlap with arboreal squirrels leads us to suggest that climbing trees even occasionally might require adequate stabilization of the eyes' position. The terrestrial

*Rhinosciurus laticaudatus* has been observed very rarely in trees[57], while the other two terrestrial rodents, *Xerus rutilus*, and *Lariscus insignis* are not tree-climbers but exhibit similar hoarding behaviours normally associated with tree squirrels[58,59]. The petrosal lobule may also have a role in visually guiding the arm movement to grasp and reach[60,61]. Therefore, this function might be useful in the case of complex foraging strategies such as hoarding food (e.g., seeds) and may explain the large petrosal lobules present in non-climbing species. Ultimately, an association between locomotor behaviour, diet and foraging strategies may impact the size of the petrosal lobules, and further testing will be required. It is worth noting that the tribe Marmotini, considered fully terrestrial, shows a small decrease in the proportion of the petrosal lobules as they likely evolved from scansorial ancestors. This might suggest that the pattern of decrease in these structures seen in the evolution of fossorial species also

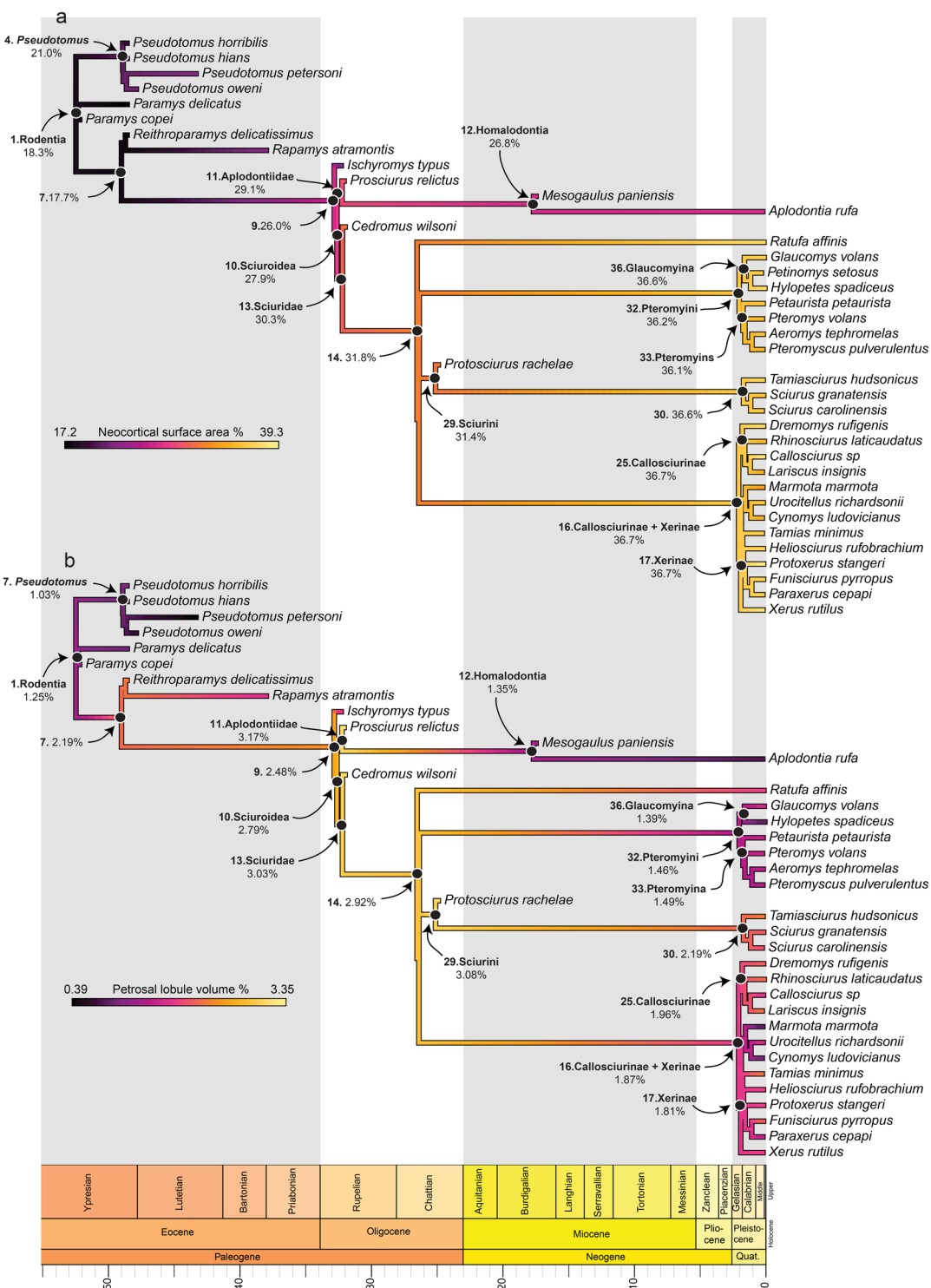

**Fig. 5 Ancestral state reconstructions with a geological timescale. a** Neocortical surface area percentage; **b** Petrosal lobule volume percentage.

occurred, to a lesser degree, during the evolution of terrestrial/burrowing Marmotini.

Flying squirrels, the Pteromyini, are among the most highly specialized rodents in terms of their locomotor behaviour. They need a substantial amount of control over their balance when gliding, but might require a different set of skills compared to arboreal taxa, because flying squirrels jump from one trunk to another and are not found on more unstable and thinner branches[62]. Our results show that gliders have relatively smaller petrosal lobules compared to arboreal taxa, but this difference disappears when body mass is considered. More specifically, small gliders have small petrosal lobules relative to body mass and brain size, while large gliders are in the range of arboreal squirrels. Interestingly, all small-sized gliders from our sample have a tail shorter than their body length, while the opposite is true for the larger gliders[63]. Tails have a role in the balance of an animal, and this could suggest a lesser need of this function in small gliders, which might also be reflected in the size of the petrosal lobules.

Our ancestral state reconstruction also finds that the greatest decrease in the proportion of the petrosal lobules during the

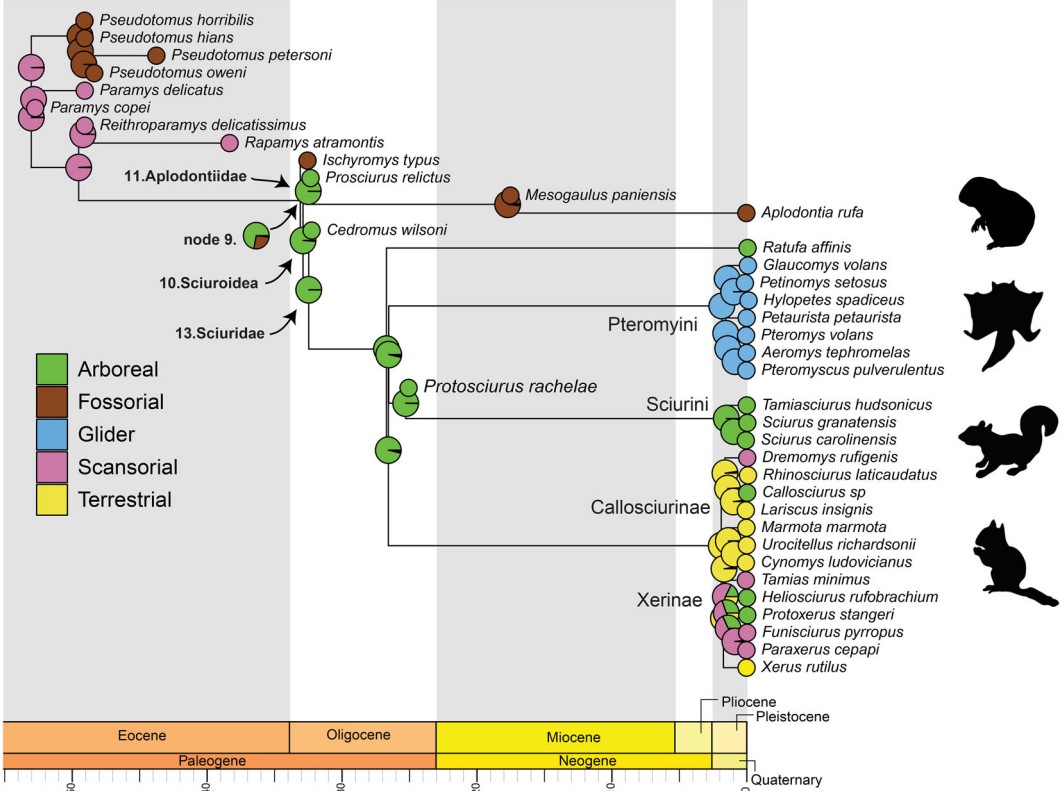

**Fig. 6 Ancestral state reconstruction of the locomotor behaviour with a geological timescale.** Each behavioural category is represented by a different colour. Gliding and terrestrial behaviours are only present in extant species, while scansoriality, fossoriality and arboreality are displayed by extinct and extant taxa.

entire evolutionary history of the rodents we studied occurred at the base of the flying squirrel lineage. Pteromyini have larger auditory bullae compared to arboreal species[64], which may reduce space for the petrosal lobules, located in the same region of the cranium. Bertrand et al.[22] noticed that the parietal region of the brain was more developed in flying squirrels, potentially suggesting enhanced hearing. Consequently, a decrease in the size of the petrosal lobules in small-bodied Pteromyini could represent an adaptation to gliding to accommodate other biological needs such as improved audition. Another possible explanation might relate to the packing problem of fitting the brain into the cranium as the latter took a more aerodynamic shape over time with the transition to gliding. As a result, the functional elements of the petrosal lobules might still be present in the non-lobule part of the cerebellum in smaller gliding species.

Previous studies using the traditional Encephalization Quotient have shown that relative brain size fluctuated through time in Ischyromyidae and Sciuroidea, with an increase in the evolution of squirrels and a decrease in aplodontiids[26,46]. It was also hypothesized that increases in the petrosal lobules and neocortical proportions occurred in early arboreal squirrels and aplodontiids as they evolved from their ischyromyid ancestors[26,46,65]. Our new study reveals a more complex pattern involving expansions and reductions of different parts of the brain, and of the brain and body mass themselves at various nodes on the phylogeny.

Along the nodes preceding the Sciuroidea node, PEQ, brain size, and body mass all gradually decreased while the opposite occurred for the neocortex and the petrosal lobules, which proportionally occupied more space in the brain compared to the olfactory bulbs. At the Sciuroidea node (~99% probability of being arboreal) a sharp decrease in body mass and small increase in brain size occurred, reflected in an increase in PEQ. At this

point, the neocortex and the petrosal lobules continued to gradually increase in proportion. This suggests that relative brain size increased as Sciuroidea became arboreal but that the proportional increase of the neocortex and petrosal lobules preceded becoming arboreal since they were already reconstructed as being expanded at the previous node. These results might imply that such neurological changes facilitated the move to the trees for rodents. Right after the split between arboreal Sciuridae and Aplodontiidae, another increase in the proportion of the neocortex and petrosal lobules took place in both clades, which is consistent with the interpretation that these traits were crucial for living in trees. However, at this point on the tree, we can already see marked differences between squirrels and aplodontiids. At the squirrel node, brain size increased, reflecting a second increase in PEQ. In contrast, the aplodontiid node shows a decrease in brain size and body mass, associated with a slight decrease in PEQ. Overall, the brain of early arboreal aplodontiids is downscaled compared to early arboreal squirrels. This suggests that very early in the evolution of Sciuroidea, squirrels, and aplodontiids started manifesting different brain organization trajectories.

In later fossorial Aplodontiidae, a shift occurred with a decrease in the proportion of the neocortex and the petrosal lobules, which, as discussed above, might reflect the less important role of these structures in animals spending more time underground. The higher PEQ at this point in the phylogeny is due to a greater increase in brain size compared to body mass. Thus, the brain was upscaled with an increase in the size of the endocast, while the petrosal lobules and neocortex became even smaller. The change in brain size might have been a response to the body mass increase, because the neocortex proportion is lower at this node.

At the node leading to extant subfamilies of squirrels, the PEQ and the proportion of the neocortex and petrosal lobules

continued to increase. The oldest known fully arboreal member of Sciurini, the extinct *Protosciurus*, had a very similar brain to that inferred for its immediately ancestral node. In contrast, modern-day Sciurini have a much higher PEQ, with the neocortex comprising a greater proportion of brain size. This means that after the Miocene, the brain of these arboreal species continued to become larger and more complex. No other fossil crania are currently available to CT scan for other still-surviving squirrel tribes, so it remains unclear if similar increases in neocortical proportions occurred before or after the Miocene in these extant groups.

Climate and temperature may have indirectly affected, or ultimately driven, neurobiological changes in rodent brains and sensory systems over the Cenozoic. During the humid, hot, tropical climate of the early to middle Eocene, scansorial, and fossorial Ischyromyidae were diverse on the northern continents[36,37,66], perhaps as a result of competition with primates, who had already seized many arboreal niches[67]. North American ischyromyids and primates started to decline in species diversity during the global cooling event of the late Eocene[38,40,67,68], which is when arboreal sciuroids appeared in the fossil record[33,39]. This environmental change may have indirectly enabled aplodontiids and squirrels to invade arboreal niches once held by primates, which by then were less diverse. Arboreal animals generally have a lower body mass compared to terrestrial species[69] and the noticeable decrease in body mass occurring in Sciuroidea at this time, may have triggered their transition into the trees. Because these body size decreases were not matched by concordant brain size decreases, PEQ increased, and regions of the brain crucial for living in a 3D environment continued to expand, with the larger neocortex indicating better vision, motor skills and/or memorization, and larger petrosal lobules suggesting improved head and eye movement control. In contrast, while many sciuroids were colonizing the canopy, the Oligocene fossorial *Ischyromys* retained a relatively smaller neocortex and petrosal lobules.

The less-dense forests and more open woodlands of the Oligocene on the northern continents, which probably played a role in the continued decline of primate and ischyromyid communities, were the stage on which squirrels and aplodontiids diversified. These more open environments were maintained during the Miocene and may have promoted the evolution of highly specialist aplodontiid burrowers[43], with their smaller brains, neocortices, and petrosal lobules. Ground squirrels were also diverse during the Oligocene and Miocene and survived past the end of the Miocene while aplodontiids waned[43] during a time when grasslands spread even further as climates became cooler and drier. Ground squirrels may have been better adapted to grasslands due to the larger neocortex they inherited from their more immediate tree-dwelling ancestors. Among their many utilities, large neocortices have been linked to sociality in primates[70], suggesting that ground squirrels may have maintained a large neocortex for a different purpose than living in the 3D arboreal environment of their ancestors.

Locomotion appears to have dramatically shaped the evolutionary history of Sciuroidea as they adapted to new environments. Squirrels and their closest relatives demonstrate the connection between locomotion and brain evolution and help us better understand how this group of rodents evolved in concert with their environment over millions of years, to produce today's astounding diversity.

## Methods

**Dataset**. We built a dataset of 42 virtual cranial endocasts, derived from high-resolution X-ray computed tomography data, from 38 species of rodents: 25 extant and 13 extinct. The resolution and other CT parameters are available in Supplementary Table 3. Our fossil sample spans the early Eocene to the early Miocene,

and incorporates all well-preserved crania accessible to us, which cover a breadth of the temporal, habitat, and locomotor diversity of these animals (Supplementary Table 4). These include 12 endocasts of well-preserved crania of ischyromyids. The sample includes 6 different species of the basal Paramyinae from the genera *Paramys* (2 species) and *Pseudotomus* (4 species) and three more derived taxa: *Reithroparamys delicatissimus*, *Rapamys atramontis* ($N = 2$) (Reithroparamyinae), and *Ischyromys typus* ($N = 3$; Ischyromyinae). Our sciuroid fossil sample includes basal taxa (the squirrel *Cedromus wilsoni* and aplodontiid *Prosciurus relictus*) and more derived species (the squirrel *Protosciurus rachelae* [$N = 2$] and aplodontiid *Mesogaulus paniensis*). We note that, for extinct aplodontiids, there are additional taxa that potentially could be sampled in the future, but these were not available to us, and it is unclear if they are well enough preserved for CT scanning (e.g., the basal *Haplomys liolophus*, the intermediate *Meniscomys uhtoffi*, and the more derived *Pterogaulus* sp., pictured in Hopkins[7]). With that said, our sample incorporates taxa that represent diverse locomotor conditions in both Sciuridae and Aplodontiidae, across a timeframe of 36 million years, which should provide an adequate framework for testing the effects of locomotor transitions on the brain through time.

Our extant sample includes 24 endocasts of squirrels and one of *Aplodontia rufa* (the mountain beaver). This sample includes members of all extant squirrel subfamilies except the monospecific subfamily Sciurillinae. Our endocasts include 10 Sciurinae (tree and flying squirrels), 4 Callosciurinae (Asian squirrels), 1 Ratufinae (*Ratufa affinis*), and 9 Xerinae (ground squirrels and African tree squirrels) species, thus spanning a range of habitats and ecologies.

**Locomotor categories and body masses**. Our dataset contains species that can be assigned to five distinct locomotor classes. Fossil ischyromyids and sciuroids were put into three main categories: scansorial (*Paramys*, *Reithroparamys*, and *Rapamys*)[36,37,71], fossorial (*Pseudotomus*, *Ischyromys*, and *Mesogaulus*)[33,72,73] and arboreal (*Cedromus*, *Protosciurus*, and *Prosciurus*)[71,74] based on a combination of postcranial and inner ear (semicircular canal) data, which are well validated proxies for locomotor behaviour[75,76]. For the extant sample, we gathered behavioural information from diverse sources[45,77–84] (see Supplementary Table 5). To differentiate terrestrial from fossorial forms, we followed Samuels and Van Valkenburgh[75]. We used their definition of semi-fossorial for terrestrial and followed their definition for fossoriality. Specifically, they define fossorial rodents as spending most of their lives underground, including foraging and building extensive burrows, while terrestrial (semi-fossorial) taxa may dig burrows but do not forage below ground. We have a total of 7 gliding, 7 arboreal, 4 scansorial, and 6 terrestrial species for squirrels; *Aplodontia rufa* was classified as fossorial. Body mass estimates were obtained using the cranial length equation from Bertrand et al.[85] for all specimens except for *Paramys copei*, for which the cheek-tooth area equation was used instead[85].

**Segmentation and endocranial quantification**. The majority of the virtual endocasts have been published[22,26,46,65,86,87] and are accessible through Morphosource (https://www.morphosource.org/), with a few exceptions. We re-segmented the rostral section of the *Reithroparamys delicatissimus* (AMNH 12561) endocast to more accurately measure areas that were previously missing from the reconstruction. A new labelfield module and 3D model of the endocast were created for this specimen (see Supplementary Fig. 3a). Additionally, we made a new estimate of the surface of the neocortex for *Prosciurus relictus*. Because the confluence of sinuses was not visible due to preservation, we used another specimen belonging to the same genus, *Prosciurus* aff. *saskatchewaensis*[26], in which the caudal margin of the neocortex is preserved to ensure accuracy in our estimation. There are also three terrestrial Xerinae species that we present endocasts here for the first time: *Marmota marmota*, *Urocitellus richardsonii*, and *Cynomys ludovicianus* (Supplementary Figs. 3b, c, and d). These were scanned at the Integrative Quantitative Biology Initiative (McGill University, QB in Canada) in 2018 and segmented with Avizo Lite 2019.4[88] using the same method as for the published virtual endocasts[46]. The "magic wand" tool was used to close any gaps between the adjacent pieces of bone and the endocranial cavity was filled to obtain a 3D model of the endocast. The volume of the olfactory bulbs was estimated by using the "volume edit" tool. For the neocortical surface area, a surface of the endocast was first generated with the "generate surface" tool. Then the region of interest was isolated with the "drawing" tool using the orbitotemporal canal laterally, the circular fissure anteriorly and the confluence of the sinuses dorsally as defining landmarks for the neocortex (Fig. 1) following previous studies (e.g.[86,89,90]). We included the superior sagittal sinus in our estimation of the neocortical size. Half of the neocortical surface was isolated, and its area doubled for *Pseudotomus petersoni* and *Reithroparamys delicatissimus* because of limited preservation. To obtain the volume of the petrosal lobules, we created a new material in the segmentation editor and isolated them from the rest of the endocast using the anterior semicircular canals to delimitate them from the rest of the cerebellum (Fig. 1). In rodents, the petrosal lobules encompass the entire paraflocculi, therefore we considered them to accurately represent the functional volume of these structures[91,92]. Because we had three endocasts for *Ischyromys typus*, two for *Rapamys atramontis* and two for *Protosciurus rachelae*, we averaged the endocranial measurements for these three species. For the previously published *Xerus rutilus*[22], the volume of the olfactory bulbs and petrosal lobules and the surface of the neocortex were obtained for the first time (Fig. 1).

**Phylogenetic tree**. The phylogenetic relationships among Ischyromyidae and with other rodents are still under debate[93–96] and there is currently no published phylogeny that includes all specimens in our sample. Paramyinae (*Paramys* and *Pseudotomus*) is usually placed as the most basal group[35,36,66,95,96]. The genus *Rapamys* has been classified as belonging to Reithroparamyinae[97] so we placed it as sister-clade to *Reithroparamys*. Ischyromyinae (*Ischyromys*) appears later in the fossil record and might be derived from a reithroparamyine ancestor[66]. Aplodontiidae and Sciuridae have been found to be closely related in molecular studies[32,98]. Based on morphology, *Cedromus* is considered a basal member of Sciuridae and as the sister-taxon to all extant squirrel subfamilies[99] and so *Protosciurus* is placed in the Sciurini tribe[100]. The aplodontiid part of the tree is based on a published phylogeny of the family[33]. *Reithroparamys* is the only taxon with an uncertain position. In a study from Meng[35] based on basicranial morphology, *Reithroparamys* was found to be closely related to Sciuroidea, while in a more recent study by Asher et al.[96], it was placed as the sister-group to *Paramys*. *Reithroparamys* has an ossified auditory bulla and common canal for the facial nerve and the stapedial artery, which are present in squirrels but not in Paramyinae[35,65,93]. Therefore, we used the Meng phylogeny as our primary tree. However, we ran the analyses using both topologies and recovered only minor differences in the results.

We used the packages *phytools* (version 0.7-20)[101] and *paleotree* (version 3.3.25)[102] in the statistical software R (version 3.6.2)[103] and R Studio (version 1.2.5)[104] to generate two calibrated trees with the cal3 Time-Scaling method[105]. We specifically used the function "bin_cal3TimePaleoPhy" because our dataset includes fossils, and we followed the guidelines proposed and described by Bapst[106] with the creation of two different matrices for dating our phylogenetic tree (Supplementary Table 6). The age ranges of the fossil specimens were based on the first and last occurrence of each species; NALMA (North American Land Mammalian Age) ages were taken from Janis et al.[107]. For the time range of extant species, we chose the age range for the species with the oldest occurrence (i.e., Pleistocene; Paleobiology Database[108]) and applied the same range for the rest of the extant sample (Supplementary Table 6) in order to avoid conflicts between intervals when using the function "bin_cal3TimePaleoPhy". The age in years of all the species was approximated by matching the narrowest geologic time bin available with the data from Vandenberghe et al.[109]. A total of 100 trees was generated and an average tree was obtained using the function "averageTree" of the package *phytools*.

**Phylogenetic Encephalization Quotient (PEQ)**. A standard practice in paleo-neurobiology is to calculate the Encephalization Quotient (EQ)[13]. This metric is used to compare the brain size of specimens with different body masses. The EQ ( $= E_i/E_c$ ) corresponds to the ratio between the actual brain size (endocranial volume here; $E_i$ ) and the expected brain size for an animal of its size ($E_c$), which is based on the following equation: $E_c = 0.12 \ (Body \ mass)^{0.67}$ from Jerison[13]. However, correlation between biological traits such as body mass and brain size might be influenced by common ancestry, especially when taxa are closely related. Thus, we used the method proposed by Ni et al.[110], the PEQ, which essentially corresponds to the same equation as EQ but is based on the phylogenetic relationships of the sample under study. To create the PEQ equation for our sample, we used the PGLS regression of the endocranial volume vs. body mass to obtain our $E_c$, which is equal to: $0.42 \ (Body \ mass)^{0.57}$.

**Permutation test and boxplots**. We used Fisher-Pitman Permutation tests to assess whether different locomotor categories had distinctive PEQs, olfactory bulb volume, petrosal lobule volume, and neocortical surface area percentages. Permutation tests are recommended over $t$ and $F$ tests when sample size is small, and can be used when the data are not normally distributed[111,112]. However, when comparing group means, homogeneity of the variances is required[111,113]. Therefore, we assessed if our data were normally distributed using a Shapiro-Wilk test and then verified the homogeneity of the variances using a Levene's test (data not normally distributed) or a Bartlett's test (data normally distributed) in R. No significant difference between the variances was found for the PEQ, olfactory bulb volume, and petrosal lobule volume percentages. The only variable for which the homogeneity of the variances was not verified was the neocortical surface area percentage. Therefore, the results of the permutation test on the neocortical surface area percentage should be interpreted with caution if group means are significantly different and the groups are visually overlapping in the boxplot. The code and detailed results from these tests are available in the code section of the supplementary information. We used the function "oneway_test" and "pairwisePermutationTest" in the package *coin* (version 1.3.1) and *rcompanion* (version 2.3.25), respectively, to perform our permutation tests and the function "ggboxplot" from the package *ggplot2* (version 3.3.0)[114] to graphically represent our results. We also provided adjusted p-values for each pairwise comparisons using "BH" also known as "fdr" correction, which controls for false discovery rate[115].

**Non-phylogenetic and phylogenetic regressions**. In order to assess the impact of locomotion, body mass, and phylogeny on the brain and its components, we ran a total of six regressions divided into two sets (Supplementary Fig. 4). We used a first set of regressions (1) to test the impact of locomotion and body mass on

endocranial, petrosal lobule, and olfactory bulb volumes. Then, we used a second set of regressions (2) to test the effect of locomotion and endocranial volume on the olfactory bulb volume, petrosal lobule volume, and neocortical surface area. In the first set (1), endocranial, olfactory bulb, and petrosal lobule volumes were considered the response variables, while locomotion and body mass were the predictor variables. In the second set (2), olfactory bulbs, petrosal lobules, and neocortex were the response variables, whereas locomotion and endocranial volume were the predictor variables (Supplementary Fig. 4). We tested two different formulas (a) and (b) on each regression. For set (1), Body mass was Predictor 1 and Locomotion was Predictor 2, while in set (2) Endocranial volume was Predictor 1 and Locomotion was Predictor 2. Formula (a) included only body mass (or endocranial volume) and locomotion as predictors in the form of Response ~ Predictor 1 + Predictor 2 while in formula (b), the interaction between Predictor 1 and Predictor 2 was also tested: Response ~ Predictor 1 + Predictor 2 * Predictor 1 (Supplementary Fig. 4).

In the first step, we tested both formulas (a) and (b) using ordinary least squares (OLS) regressions. We elected to test these relationships without taking into account phylogeny as in some instance OLS outperforms phylogenetic generalized least squares (PGLS)[116] analyses when no phylogenetic signal is present in the data[117]. We used the "gls" function and the "anova" function of the *nlme* package (version 3.1-142)[118] to select the best formula based on the lowest AIC (Akaike Information Criterion) value. Then, using the selected formula, we assessed whether a PGLS analysis that takes phylogeny into consideration[119] should be used instead of OLS. We plotted the residuals from the OLS analysis against the species ordered by phylogeny. In the event of related taxa having similar residuals, the data were considered not independent, and phylogeny needed to be accounted for and therefore a PGLS was used instead of OLS. In all cases, PGLS was chosen based on this reasoning (Supplementary Fig. 4). Therefore, we re-ran all analyses separately using PGLS.

For each of the six PGLS regressions, we chose between four different models of evolution: Brownian motion (stochastic model with changes occurring randomly[120]); Lambda model (Brownian motion with internal branches multiplied by Pagel's λ [lambda])[121]; Early Burst (model with initial high evolutionary rate, which then slows down[122]); Ornstein-Uhlenbeck (model with a selective optima, adaptative zone[123]). We used the "gls" and "anova" functions to select the best model based on the lowest AIC (Akaike Information Criterion) value. First, we tested which model of evolution (i.e., Brownian, Lambda, Early Burst, and Ornstein-Uhlenbeck) had the lowest AIC value for formula (a). Then, we did a similar test for formula (b). The two models from formulas (a) and (b) with the lowest AIC values were then compared using AIC to obtain our final model of evolution. We repeated this process for each of the six regressions. All AIC values for each equation are available in the code section of the supplementary information.

In the end, we obtained a total of six final PGLS regressions, which form our basis for assessing the relationship between the size of brain components, body mass, phylogeny, and locomotion. The three regressions for Set 1 are denoted as 1EB (assessing endocranial volume in relation to body mass and locomotion), 1PB (assessing petrosal lobule volume in relation to body mass and locomotion), and 1OB (assessing olfactory bulb volume in relation to body mass and locomotion). The three regressions for Set 2 are denoted 2NE (assessing neocortex surface area in relation to endocranial surface area and locomotion), 2PE (assessing petrosal lobule volume in relation to endocranial volume and locomotion), and 2OE (assessing olfactory bulb volume in relation to endocranial volume and locomotion).

Each PGLS regression is associated with p values, which denote whether the predictor variables (in this case body mass and locomotion for the three regressions of Set 1, and endocranial volume and locomotion for the three regressions of Set 2) significantly predict the response variables (endocranial volume, petrosal lobule volume, and olfactory bulb volume for the regressions of Set 1; neocortical surface area, petrosal lobule volume, and olfactory bulb volume for the regressions of Set 2). We also report ʎ obtained from running the Lambda model for each regression, which indicates the degree of phylogenetic signal existing for a specific model[121]. When ʎ is close to 1, the phylogenetic signal is strong, while the opposite is true when ʎ is close to 0. Additionally, we plotted our six regressions to visually estimate how each locomotor group compared to the others. We used the "ggplot" function from the package *ggplot2* (version 3.3.0)[114] to make the different plots.

**Non-phylogenetic post-hoc tests**. The PGLS regressions can test whether locomotion as a whole significantly predicts response variables (such as endocranial volume), but cannot test for significant differences between different states of locomotion (e.g., whether arboreal taxa have larger endocranial volumes than fossorial taxa)[124].

Thus, to further investigate the role of locomotion types on endocranial size and the size of brain components, we performed a non-phylogenetic ANOVA based on the Residual Randomization using the function "lm.rrpp" and ran post-hoc tests with the function "pairwise", which are both from the package *RRPP* (version 0.6.1). While these tests make comparisons between locomotor categories, they are not phylogenetically corrected and should be interpreted with caution. We could not use the function "phylANOVA" from the *phytools* package that takes phylogeny into account because it only allows testing for one predictor variable and

we had two (i.e., locomotion and body mass). Another option would have been to use the residuals of the relationship between endocranial volume and body mass to incorporate body mass. In this case, locomotion would be the sole predictor variable and the function "phylANOVA" could have been used. Even though this would have been a solution for 1.OB, it would not have been useful for regressions 1.EB and 1.PB, which incorporate a third predictor: the interaction between body mass and locomotion. As such, because we wanted our various analyses to be maximally comparable, we opted not to use this function. All data were converted to $\log_{10}$ prior to the analysis and all regressions were produced in R Studio with the packages *ape* (version 5.3)[125].

**Ancestral character state reconstruction**. We produced ancestral state reconstructions using maximum likelihood (ACSRML) for continuous characters (endocranial volume, olfactory bulb, and petrosal lobule volumes, neocortical and endocranial surface areas, body mass, PEQ, and residuals) using the function "fastAnc" in the package *phytools* (version 0.7-20)[101]. Because *Petinomys setosus* lacks the petrosal lobules, two different time-calibrated trees were used for the reconstructions: one for the petrosal lobules and one for the rest of the variables (Supplementary Fig. 5; see associated node labels in Supplementary Table 2). We also performed an analysis to reconstruct the ancestral states of locomotor behaviour. We used the function "make.simmap" from the same package using the Q = "empirical" option, which fits a continuous-time reversible Markov model for the evolution of the discrete values of the tips (i.e., locomotor categories), simulating stochastic character histories employing that model and the states on the tips of the tree[126–128]. We used a transition matrix with an equal rate model, which assumes equal rates of transition between different states (i.e., locomotor categories), and without assuming any directionality of change between states. We chose this model because we decided it was more appropriate to not assume any differential rates or specific directional changes between different locomotor categories. We performed 1000 simulations of stochastic character maps that we then summarised in a unique plot. We used the function "geoscalePhylo" from the package *strap* (version 1.4)[129] to add a geological time scale to our results.

**Reporting summary**. Further information on research design is available in the Nature Research Reporting Summary linked to this article.

## Data availability
Previously published endocasts[22,26,46,65,86,87], the new surface rendering of the endocast of *Reithroparamys delicatissimus* and those of the three extant terrestrial squirrels generated for this project are available in MorphoSource (www.morphosource.org[130]) at https://www.morphosource.org/Detail/ProjectDetail/Show/project_id/83.

## Code availability
The code used to conduct the different analyses is available in the code section of the supplementary information and at the following GitHub repository: https://github.com/Bertrand-Ornella/Brain-evolution-Sciuroidea.

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

## Acknowledgements

We thank Dr. Hans Larsson from McGill University in Montreal (Quebec) for his generous help and for facilitating the scanning process of the three new squirrel specimens, and Rui Tahara for scanning the specimens at the Integrative Quantitative Biology Initiative (McGill University, QB). We also thank the American Museum of Natural History (AMNH) Mammalogy collection team in New York for sending the specimens to Montreal for them to be scanned. This research was funded by a Marie Skłodowska-Curie Actions: Individual Fellowship (H2020-MSCA-IF-2018-2020; No. 792611) to OCB; a European Research Council (ERC) Starting Grant (No. 756226) to SLB, under the European Union's Horizon 2020 Research and Innovation Programme; a Philip Leverhulme Prize to SLB; the National Agency for Research and Development (ANID)/ PFCHA/Doctorado en el extranjero Becas Chile/2018-72190003 to HPP; a Leverhulme Trust Research Project grant (RPG-2017-167) to PI SLB, which funds JAS; an NSERC Discovery Grant to MTS.

## Author contributions

OCB conceived and designed the study. OCB and MTS acquired the dataset. OCB and SLB wrote the manuscript. Analyses were performed by OCB, HPP, and JAS. All authors carried out interpretations, revised the manuscript, and provided final approval before submission.

## Competing interests

The authors declare no competing interests.
