## [Peer Review File · Communications Biology]

Reviewers' comments:

Reviewer #1 (Remarks to the Author):

NB With the consent of the handling editor, this review has been jointly produced between myself and my PhD student.

This is an interesting study on brain size and shape evolution in squirrels and aplodontiids, as revealed through virtual endocasts. The study reveals changes in the relative size of the brain through time in this clade, as well as changes in the relative importance of several regions of the brain. In general, the manuscript is written well with very nice figures, and the background to the study is covered well. However, we have some queries regarding the methods and a note of caution regarding the interpretation of results.

Our major concern is with the statistics outlined on page 11. You have performed a linear regression for each of the six relationships you wish to test, but then you go on to say "we added each locomotor type to test their individual effect on each relationship". I'm afraid we don't understand what 'adding each locomotor type' means. Table 1 suggests that each locomotor category was tested separately, yet the total number of specimens in the analysis remained the same, so we're not sure how this test would work. Would this not be better conducted as an ANCOVA with locomotor type as a covariate? This would test for differences in the relationships between your different locomotor categories. You could then use post-hoc tests to work out precisely which locomotor categories differ, if and only if the ANCOVA is significant. This would avoid your current method of splitting the locomotor types off one by one, which runs the risks associated with multiple comparisons.

If you would rather stick with the method as it currently stands, you're going to need a much clearer explanation of how the tests were conducted and some references to previous work that has carried out tests in this way before. In addition, you'll also need clearer explanations of what all the different parameters and their associated p values mean. Table 1 is currently a wall of data and it's very difficult to pick out what's important and how these numbers relate to the conclusions you're drawing. You've currently reported the slope and intercept, but why are these interesting? What do they tell us about the relationships between your variables? And what do the associated p-values represent – significant difference from zero, or from those parameters in other analyses? And lastly, what does 'significance of the regression' mean? Do you mean significance of the R-squared value?

Our other main point is in regard to conclusions. One of the main take-aways from the study is that fossorial species have relatively smaller brains and arboreal species relatively larger brains. However, it's very difficult to tease apart the effects of locomotor strategy and size here. According to the craniofacial evolutionary allometry hypothesis (CREA) larger mammals tend to have smaller braincases and longer rostra. In your sample here, almost all of your fossorial species are large and most of your large species are fossorial. We don't think there's much you can do about this in terms of your sample, but it would be good to see an acknowledgement in the discussion that, although the patterns seen here may be related to locomotion, there are other possible explanatory factors.

We have more specific comments listed below, line by line:

Lines 26-29: The last two sentences of the abstract seem unrelated to the study here

Line 33: Sciuridae is the third-most diverse rodent family (after Muridae and Cricetidae)

Line 36: An explanation of the distinction between fossoriality and terrestrial mammals that burrow would be very helpful here. What distinguishes Aplodontia as fossorial compared to ground squirrels?

Line 37: There seems to be growing agreement that the family name of mountain beavers should

be spelt 'Aplodontiidae' (with two Is). It's spelt this way in the Handbook of the Mammals of the World.

Line 53: There is also considerable radiation of murids in Miocene which may have contributed to aplodontiid decline.

Lines 83-85: The numbers don't add up here.

Line 84: What was the resolution of the scans?

Line 88: For some fossil species, you had multiple specimens from which you took a mean. Did you also try including them all as individuals in the dataset, to get an idea of the extent of intraspecific variation?

Line 109: Were the extant species assigned locomotor categories on the basis of ear morphology or from published behavioural studies?

Line 114: This is a strong statement that underpins your analyses. It needs referencing.

Line 185: This list of variables is in a somewhat random order. Maybe list from largest to smallest?

Line 209 "For each relationship, we obtained two p-values, one for the slope (always zero) and one for the intercept, providing information about the significance of the relationship." Do they really? Surely, the slope and intercept tell you something about the relationship between the two variables, and it is the R-squared value that indicates the strength of the relationship. And why is the p-value of the slope always zero?

Line 236: A permutation test is a better way (than Kruskal-Wallis) of assessing significance between groups in which the data isn't normally distributed

Line 238: Why was phylogeny not considered here? Was the phylogenetic correction within PEQ considered sufficient to account for evolutionary history?

Line 248: 'simulation' should be 'simulations'

Line 277: Tell us in which case the residuals were not normally distributed

Line 277: The list of significant regressions with all its 'and' and 'vs.' can get very confusing to read. Is there a clearer way of presenting this? A similar thing happens at line 292.

Line 294: The three consecutive adverbs in this sentence make for a head-spinning read. Could it be phrased a little more simply?

Line 294: Are the petrosal lobule and neocortex larger relative to body mass or endocranial volume?

Line 300: The boxplots seem like a key part of your results. It would be nice to see Figure S5 as part of the main manuscript rather than in the supplementary info.

Line 305: It would be very useful to have a version of Figure 2 with all the different locomotor types represented, so that the reader can see how they all compare to one another, not just the ones you went on to analyse further.

Line 323: To clarify, PEQ increase in Pteromyini was related to both a brain size decrease and body size decrease?

Line 333: 'surface area' rather than 'volume'

Line 337: Do you mean that the neocortex expanded IN Sciuridae and Aplodontiidae, or should that be on the lineages leading to?

Line 365: 'Sciurini' rather than 'Sciurine'

Line 503: change 'to what is inferred' to 'to that inferred'

Line 507: Are there other fossil species out there that would help resolve this question? What would you liked to have scanned in an ideal world?

Table S4: Minus sign missing from first number?

Philip Cox
Jesse Hennekam
University of York

Reviewer #2 (Remarks to the Author):

Thank you for asking me to review the manuscript entitled "The impact of locomotion on the brain evolution of squirrels and close relatives during the Cenozoic". This study incorporates several

advanced phylogenetic approaches to present a thorough analysis of ~50 million years of brain evolution and ecological diversification among squirrels and relatives. It incorporates endocast metrics and phylogenetic data from both extant and extinct species and presents the results in visually pleasing figures. The study also introduces three entirely new endocasts from *Marmota marmota*, *Urocitellus richardsonii* and *Cynomys ludovicianus*, which are all nice additions. They have discerned significant ecomorphological trends in brain shape and estimated probabilities of potential ancestral states across various brain regions and niches. I enjoyed reading this manuscript and feel that it has much to offer. However, it still requires some substantial reformatting of the narrative and clearer justification for some of the methodological choices that have been made.

MAIN POINTS

1) A key problem with the arguments presented here is that the authors routinely conflate locomotion and overall ecology, the most obvious example being in the title itself. Locomotion strategies are even listed as "Ecology" in Table S1. While locomotion is undoubtedly a defining aspect of each niche represented by the taxa included, it is highly unlikely that locomotion alone is responsible for all their findings. Interestingly, a large chunk of the discussion is committed to pointing out other selective pressures of different niches which may contribute to the findings of endocast shape, such as communication, vision, and audition, but there are many others, including foraging strategies, predator avoidance, and courtship behaviours, which are all likely more complex in arboreal environments than they are underground. While the authors present a fairly sound case for the petrosal lobules having a clear association with locomotion, neocortical surface area is a more difficult sell. For example, in gliders the petrosal lobules are relatively smaller while the neocortex remains consistent with arboreal species (Figure 4). It seems to me this study is more associated with the effects of niche diversification on the brain in general, rather than locomotion alone. Teasing apart the contrasting conditions of each lifestyle would provide a more complete picture than what has been presented in the manuscript's current form.

2) Abstract:

A little more detail on what kind of data was collected from the endocasts and what types of tests were carried out would help here.

3) Introduction:

There's some great points in this section. But it is short, at only three paragraphs, which undersells the significance of the work that has been put in and the results they present. There also seems to be a lack of clear direction. The paragraphs are a little disjointed and missing a clear rationale for conducting this study. The first paragraph details rodent/squirrel diversity, the second describes their evolutionary history. This information is of course important, but this is fundamentally a study on brain morphology and its association with ecological diversification through evolutionary time. Endocasts and their potential to resolve such associations are not mentioned until the third and final paragraph, and the methods used aren't specifically mentioned at all. More context should be brought to what is known about brain ecomorphology to date, and sooner in the Introduction. How is body size associated with brain size in these animals? And in mammals in general? Is there modularity evident in brain allometry? What features of the brain are associated with the specific selective pressures of different niches? How do these features differ between environments? And across climates? Do these trends transcend any modularity of the brain associated with allometry? And are these brain/niche correlations consistent across mammalian taxa? Can a climatic influence on the evolution of brain shape therefore be discerned? How can endocasts be used to answer these questions? Most importantly, what is expected to be found from these analyses? And how could the findings be expected to improve our understanding of mammalian brain evolution in general? Some of these questions may be easy to answer with the current literature, some possibly less so. But as it stands, the authors have only briefly touched on them in a couple of sentences and have only provided two citations - both of which are their own previous work. There isn't a clear hypothesis presented, but rather the reason for the

study seems to be simply because these tests haven't been done on these animals before. I believe a stronger argument can be made than that.

Methods:

4) The authors have presented an extensive assortment of analyses, tables, and figures, both in the manuscript and supplementary. The first thing that confused me was the number of specimens used. There are 44 endocasts made from 38 species of rodent: 28 extant species and 13 extinct species. But $28 + 13$ is 41, not 38. And are some species therefore represented more than once to reach $n = 44$? But then the number of species presented in Table S1 is 38, so I'm not following. Then it is stated that the data set includes 14 endocasts of well-preserved crania of ischyromyids. Aren't these all extinct? If so, is it 13 extinct species used, as stated earlier, or is it 14? I trust that the authors know what's going on here, but I think a reduced version of Table S1 would be immensely helpful here, denoting the number of individuals of each species, their body mass, habitat/locomotion, and their extant/extinct status.

5) I have some concerns about measuring volume of the petrosal lobules from endocasts. Endocasts do not contain information on internal brain anatomy so it can be tricky to get reliable estimations of volume from some brain sections. While the olfactory bulbs are fairly easy to recognise as an isolated mass protruding from the brain, the petrosal lobules are less obvious in some species. So how is the functional volume of these structures below the surface morphology of the endocast reliably estimated? The authors even suggest in the Discussion that the observed smaller sizes of the petrosal lobules in the Pteromyini may be a result of them being obscured in some manner by the auditory bullae or distributed differently due to packing of the brain. If so, how can we know if they are in fact smaller or not? They may just be structured or positioned differently.

6) The authors present two phylogenies for their sample, "Meng" and "Asher" and choose to present "Meng" in the manuscript and "Asher" in the supplementary. But no reasoning for this choice is provided. Was it arbitrary? They state that only minimal differences were found. If so, the supplementary figure probably isn't required at all if the minor discrepancies are detailed in the manuscript for justification.

7) I admit that I have limited experience with some of the more advanced phylogenetic regressions carried out. All tests make perfect sense to me though, at least up until the authors add each locomotor type to test their individual effect on the phylogenetic tests. The first part that concerns me is that each individual type of locomotion is tested for the most appropriate model via the Akaike's Information Criteria. I don't think I've ever seen all groups within a single discrete variable be individually tested for their own most appropriate model. Shouldn't each type of locomotion be standardised to be tested under the same model? Otherwise there may be a risk of over-manipulating the data, especially for such small sample sizes for each group. Are there other studies that have done this previously? I think, at minimum, the authors need to be clearer about what they are doing here and justify their decisions with citations of prior studies that employ similar methods when possible. The second part of this I'm unsure about is how sample size (N) can be consistent across each locomotion type, at $N=37-38$, when all locomotion groups are $N \leq 10$. The Terrestrial group only has an N of 6. So how is Terrestrial locomotion tested across all 38 species? Is it because each locomotion strategy is treated as a binomial presence/absence for each model? Wouldn't it make sense (and be easier) just to test locomotion as a categorical variable in the initial model and carry out pairwise comparisons between locomotion groups? I would like to see this section made a lot clearer.

8) It is a little strange that there is a paragraph detailing the generation of boxplots that are only presented in the supplementary. All methods in the main manuscript should be relevant to all results presented in the main manuscript. Many people would likely read how these boxplots were created and will never see them. The authors should probably consider whether to include the boxplots in the main manuscript or not at all.

Results and Discussion:

The Results section is great. Nice figures and findings that are easy to follow. The Discussion could use some restructuring to follow on from my earlier suggestions regarding the Introduction, so I'll

leave that for now. But two things are worth a mention.

9) Firstly, the authors present temperature data and oxygen data in Figure 3, modified from Zachos et al. In the Discussion, they then mention the potential significance of climate in the evolution of these animals. However, there is no context provided prior to Figure 3 to indicate that climate was ever an important consideration of their study design. Climate and climate variables aren't mentioned to any degree of importance in the Introduction, nor mentioned at all in the Methods.

10) Secondly, a minor thing to address here is that, while these tests do demonstrate clear significant associations, they are still based on results that are limited by the diversity of the sample and variables tested. The authors need to make more careful use of language that acknowledges this point in both the results and discussion. Various events are sometimes presented as fact, rather than interpreting their results as supporting specific hypotheses. Results will always be limited to the data analysed (in this case, 38 out of more than 300 known species of squirrels and relatives), and so definitive statements should be kept to a minimum, particularly regarding reconstruction of ancestral states, which are based on probabilities generated from the taxa used. For example, the results SUPPORT the existence of an arboreal ancestor for Aplodontidae and Sciuridae. The results can't, and don't, simply demonstrate that their ancestor WAS arboreal. I'm sure the authors are aware of the difference and fully intend for the former interpretation to be received, but it's a minor linguistic detail that can lead to an incorrect understanding by readers.

As I stated, I enjoyed reading this study. It represents a thorough analysis of the ecomorphology and evolution of the brain in a very charismatic group of mammals. I'm confident that the authors can address the aforementioned points and I hope that they are received in the constructive manner that is intended. I look forward to seeing the revision.

MINOR COMMENTS

11) Line 23. "We find that arboreal species have larger brains with bigger neocortices and petrosal lobules compared to fossorial taxa"
Absolutely or relatively?

12) Line 25. "which may have facilitated the move into a complex three-dimensional environment"
Aren't all environments 3-dimensional? Suggest rewording. Dynamic perhaps?

13) Lines 33-34. " Squirrels (Sciuridae), the second most diverse extant family of rodents..."
According Burgin et al., (2018) the Sciuridae are the third largest rodent family, behind Muridae and Cricetidae.

14) Line 37. Remove "hugely".

15) Line 38. How diverse were the Aplodontidae?

16) Lines 38-40. "The striking differences between living sciurids and aplodontids raise the question of how the modern rodent assemblage emerged."
This sentence hints that this study is addressing questions at the Rodentia scale. But it isn't. It's confined to sciurids and close relatives.

17) Lines 41-42. "The likely common ancestors of squirrels and the mountain beaver, known as ischyromyids, were probably fossorial to scansorial."
This is an odd way to start the paragraph, because there has been no prior context provided as to why locomotion is important in rodent evolution.

18) Lines 50-51. "...Then, in the drier and more open environment of the Miocene (23-5.3 million

years ago), aplodontids shifted to become predominantly specialized diggers, while squirrels continued to occupy the same Oligocene niches and added gliding to their repertoire.”
I’d be careful with this word usage. No niches are the “same”. Also, it’s probably more correct to say that gliding evolved in some species, rather than the whole taxon “adding” gliding.

19) Lines 65-66. “...the relationship between the form of the brain and ecology has yet to be assessed in these rodents over time, in a phylogenetic context, to test whether different locomotor groups had uniquely different brains and how these evolved.”

Is this the hypothesis? How are they expected to differ? What is expected to be teased apart from phylogenetic shape differences between modes of locomotion/ecology? What impact will the findings be expected to have on our understanding of mammalian brain evolution?

20) Lines 66-68. “Here we use a CT scan dataset of extinct and extant squirrels, aplodontids, and close relatives, and assess changes in the brain over time, across phylogeny, and associated with ecology.”

How are you planning to do this? Any interesting and/or unique methods? These could be briefly mentioned here.

21) Lines 75-80. Figure 1 looks nice. But most of it is not relevant, nor referenced in the text. Suggest instead to include just three angles of part C (dorsal, lateral, and ventral). Also, there is no ‘e’ in the Figure. All parts need to be labelled.

22) Line 108. A Table would help so much here. See previous comments.

23) Line 114. Citation needed after “...which are well validated proxies for locomotor behaviour.”

24) Lines 185-187. “For each taxon we obtained six variables: endocranial volume, body mass, olfactory bulb volume, petrosal lobule volume, neocortical surface area and endocranial surface area.”

Some of these have already been mentioned. Suggest moving this passage up to before any methods for measuring them have been described.

25) Line 262. Table S5 is first mentioned after Table S6. Their names should be reversed.

26) Lines 265-268. I’m not following everything going on in Table 1. Each of the p-values need to have a separate heading.

27) Lines 272-272. “The models with the lowest AICc values were chosen as the best-fit model for each analysis (Tables S4, S7).”

This sentence is not needed here. Should be mentioned in the methods already.

28) Lines 273-275. “Phylogenetic generalized linear squares models (PGLS) always provide a better fit to our data than the linear squares regression (OLS), except for one regression using Meng’s topology (see Table 1).”

OLS is mentioned in Table 1 but PGLS isn’t. The Table should have both PGLS and OLS defined to avoid confusion from this sentence.

29) Lines 306-310. Have these regression lines been generated for each group independently? I find it hard to believe they would all be perfectly parallel – particularly Fossorial in part D.

30) Lines 359-360. “The locomotor behavioral ancestral state reconstruction showed that the common ancestor of rodents was scansorial (Fig. 5).”

MOST LIKELY scansorial, based on your results.

31) Lines 379-381. “The PGLS and OLS regressions show that arboreal species have significantly

larger brains, petrosal lobules and neocortices compared to other locomotor modes..."
Relatively or absolutely?

32) Line 422. I don't think the research cited here states this correlation.

33) Lines 432-434. "The ancestral state reconstruction of the Marmotini tribe (ground squirrels) shows a small decrease in the proportion of the petrosal lobules and neocortex..."

I can see it in the petrosal lobules and PEQ but not the neocortical surface area in Figure 4. Am I missing something?

34) Lines 467-469. "Our new study reveals a more complex pattern involving expansions and reductions of different parts of the brain."

Good opportunity to discuss evidence for modularity here.

Reviewer #3 (Remarks to the Author):

The present manuscript evaluates the role of locomotion (scansorial, fossorial, gliding etc.) in shaping the evolution of the brain in rodents (squirrels) over several millions of years. The authors use CT scans and virtual endocasts of the braincase to reconstruct the size of portions of the brain in fossil and extant squirrels. They then analyze the relationships among locomotion, body mass, and the size of various regions of the brain as well as use ancestral character estimation to reconstruct the likely ancestral states of brain characteristics and body size. The authors also form some hypotheses relating changes in global climate during the period of squirrel evolution to the appearance of new locomotor types and diversification within those groups. This is the first study I am aware of to combine fossil and extant squirrel species into an analysis of brain size. The analysis also covers a very large portion of the Cenozoic, also making it the most temporally expansive study of its kind, to my knowledge. In this sense, the paper is quite fantastic.

I am primarily concerned about the statistical analyses for two reasons 1) a lack of clarity regarding how the models were actually fit, 2) I don't believe the authors are really testing the hypotheses they think they are (see below).

My comments on the methods and results will reflect a general confusion I had that stemmed from a lack of clarity in the text. I only figured out (maybe?) how the models were fit after looking at Figure 2. This is insufficient. The text should say "we performed these univariate regressions for all taxa and then for subsets of taxa within each locomotor group."

But then this brings me to my second concern regarding the hypotheses they are testing. Throughout the discussion, in particular, the authors state things like "locomotor mode had a significant impact on relative brain size etc." But most of the analyses they use do not actually test this (in fact, only the pairwise comparisons of box plots does). What they actually test with the regression analyses is whether the relationships between brain morphology and body mass etc. vary among groups with different locomotor modes. Based on the questions set up in the manuscript, I can't understand why the authors would opt for this approach over a multiple regression approach that simultaneously tests for differences in brain morphology among groups and say brain morphology and body mass.

Perhaps, my confusion simply stems from the fact that their hypotheses are not clearly communicated at the outset of the paper. In which case, this is an easy fix with little to no re-analysis. But based on the way things are worded in the discussion, I am uncertain.

Introduction

Paragraph starting on line 57 – The authors don't present any citations for their interpretations of the functions of the different brain regions. I think this is an oversight (and, in fact, the citations I was looking for show up in the discussion. They should probably also be cited in the introduction).

The introduction does not present any particular hypotheses. This may really have helped later on when I was trying to discern why the authors fit the models they did. It might also help parse out to what degree the paper is focused on correlating brain morphology and ecology and change in brain morphology through time. It seems to be both but the introduction almost makes it seem like the paper will be more focused on the time component.

Methods

Although the sample size is small relative to the diversity of rodents, I acknowledge the time and resource intensive nature of CT scanning and creating the virtual endocasts used here.

BUT I do wonder about the statistical power of their analyses. There are very few samples per locomotor group, and so the power to detect differences (or to perform regressions on subsets of the data) may be particularly low (though, the fact that they do detect differences probably means that more samples would only strengthen the relationship).

Line 114 – states that there is a validated relationship between inner ear/post-cranial morphology and locomotor behavior but no citations are provided.

Line 119 – no citation provided to support using cheek tooth area as a predictor of body mass

The method of producing phylogenetic topologies and time calibrating them is sound, to my knowledge.

Phylogenetic regressions – This is where I start to get a bit confused about how the authors set up their regression analyses. I think this stems from the difference in how I understood it was done from the text, how Table 1 is structured, and how Figure 2 is presented.

From the text, I assumed that univariate models were fit (e.g. endocranial volume vs body mass) and then multivariate models were fit (e.g. endocranial volume vs. body mass and locomotor mode).

However, table 1 suggests that the authors only employed regressions with one independent and one dependent variable. What is the hypothesis being tested here? From the introduction, I thought the main goal of the manuscript was to test for a relationship between ecology and brain morphology/body mass. So, univariate regressions don't seem like the right approach to me.

I would think they authors would need to use multiple regression. For example:

Endocranial volume \sim body mass + locomotor behavior

Or

Endocranial volume \sim body mass + locomotor behavior + body mass:locomotor behavior (with this representing the interaction between the two variables)

These would simultaneously test for the relative effects of body mass and locomotor behavior (as well as any interaction between the two because, for example, arboreal rodents don't tend to be as large as marmots).

But things get more confusing on Line 191, where the authors suggest that they did add locomotor

behavior to their regressions. But coming back from looking at Figure 2, it seems that the authors performed regressions for all of the species and then just for subsets of species from particular locomotor modes. How are these regressions relevant to the questions set forth in the study? Wasn't the goal to look for a relationship between ecology and the brain? These test whether the relationship between the brain/body mass etc. are different among groups with different ecologies. A different question entirely.

The authors could have simply done step wise model fitting, picked the best model based on AIC, and run with that rather than first fitting all of the univariate combinations.

Line 193 – keep in mind that the strength of a relationship is only indicated by r^2 or other effect sizes and not p-values.

The methods and hypotheses need serious clarification. The way the models were fit is exceedingly unclear until you get to Figure 2, then it was like an "AH HA" moment.

Line 199 – the authors mention they compare the fit of models assuming different correlation structures among the variables but say nothing about which variables actually went into these models.

This also seems like a very non-traditional approach to selecting between phylogenetic and non-phylogenetic regressions. Typically, one would test for phylogenetic signal in the residuals for the OLS and then implement the phylogenetic regressions. That being said, I don't think the approach is necessarily wrong.

Table 1 – The caption needs to be clarified. In combination with the lack of clarity in the text, I did not understand why the table had 6 parts until my "eureka moment" with Figure 2.

Line 284 – The authors use very non-standard statistical terminology here. "Three lifestyles have an impact on specific relationships." What does this mean? I now deduce that they mean that the relationship between brain/body mass differs among species from the different locomotor groups. Please simply say that. The way it is stated now is simply incorrect and misleading.

In general, all of the wording with "when locomotor mode was included" is misleading. They did not include locomotor mode in the analyses. They analyzed subsets of the data for relationships among brain components/body mass.

Line 294 – why do these analyses come after the regressions? They seem to set up some basic information that a reader might want to know before diving into the model results.

Figure 2 – The caption needs to describe why they were two regression lines. I now understand that one is the regression of all species? And the other is just the subset from a particular locomotor group? Please make this clearer.

The power of some of these individual locomotor mode regressions must be extremely low.

Figure 3 – the raw data are available as part of the Zachos et al. (2008) paper.

Figure 3 and 4 are really nice (except the Zachos curve). Why not put the cute animals on Figure 3 and 4 as well?

Line 379 – an impact on the relationship between brain morphology and body mass etc. I think the way this is worded is throwing me off. It also uses the word significant in a way that I don't think it should be used. The authors aren't presenting a regression of locomotor mode on these other variables.

Line 382 – but you don't test this with the models.

– a lot of the citations are was searching for in the introduction are here.

Line 410 – but your models don't test this (only the boxplots do)

425 – significant impact, again this wording seems inappropriate

Answers to reviewer #1:

This is an interesting study on brain size and shape evolution in squirrels and aplodontiids, as revealed through virtual endocasts. The study reveals changes in the relative size of the brain through time in this clade, as well as changes in the relative importance of several regions of the brain. In general, the manuscript is written well with very nice figures, and the background to the study is covered well. However, we have some queries regarding the methods and a note of caution regarding the interpretation of results.

Thank you very much for this very useful feedback. We have taken the different suggestions into consideration. We reviewed our methods and updated our results and discussion (see below for more details).

Our major concern is with the statistics outlined on page 11. You have performed a linear regression for each of the six relationships you wish to test, but then you go on to say “we added each locomotor type to test their individual effect on each relationship”. I’m afraid we don’t understand what ‘adding each locomotor type’ means. Table 1 suggests that each locomotor category was tested separately, yet the total number of specimens in the analysis remained the same, so we’re not sure how this test would work. Would this not be better conducted as an ANCOVA with locomotor type as a covariate? This would test for differences in the relationships between your different locomotor categories. You could then use post-hoc tests to work out precisely which locomotor categories differ, if and only if the ANCOVA is significant. This would avoid your current method of splitting the locomotor types off one by one, which runs the risks associated with multiple comparisons.

If you would rather stick with the method as it currently stands, you’re going to need a much clearer explanation of how the tests were conducted and some references to previous work that has carried out tests in this way before. In addition, you’ll also need clearer explanations of what all the different parameters and their associated p values mean. Table 1 is currently a wall of data and it’s very difficult to pick out what’s important and how these numbers relate to the conclusions you’re drawing. You’ve currently reported the slope and intercept, but why are these interesting? What do they tell us about the relationships between your variables? And what do the associated p-values represent – significant difference from zero, or from those parameters in other analyses? And lastly, what does ‘significance of the regression’ mean? Do you mean significance of the R-squared value?

Lines 230-310: We extensively modified our methods so that they are tailored to testing for brain differences between locomotor categories, and so that they are clearly explained. We performed three different types of analyses: (1) permutation tests instead of Kruskal-Wallis tests. We have moved the boxplot in the main document; (2) PGLS analyses that considered locomotion as the predictor variable, rather than having a

specific locomotor mode being the variable. We also included a figure illustrating how each locomotor mode differ from one another; (3) non-phylogenetic corrected ANOVA using Residual Randomization and associated post-hoc pairwise tests to look at how locomotor groups differ from one another. The reason for using non-phylogenetic post-hoc tests stems from the fact that, as outlined in the most recent literature, post hoc tests cannot currently be done following a PGLS analysis (Weisbecker et al., 2020).

The Asymptotic K-Sample Fisher-Pitman Permutation Test and pairwise comparisons were done on the PEQ, olfactory bulb, petrosal lobule, and neocortex percentages (Table 1). We selected 6 different regressions by using the AIC index (see Tables 2, 3). We made a summary of the PGLS process in Fig. S2. The ANOVA and non-phylogenetic corrected postdoc tests are in Tables S6, and 4. The code is also available in the supplementary data and in a GitHub repository.

Weisbecker V, Speck C, Baker AM (2020) A tail of evolution: evaluating body length, weight and locomotion as potential drivers of tail length scaling in Australian marsupial mammals. *Zool J Linn Soc* 188 (1):242-254. doi:10.1093/zoolinnea/zlzl055

Our other main point is in regard to conclusions. One of the main take-aways from the study is that fossorial species have relatively smaller brains and arboreal species relatively larger brains. However, it's very difficult to tease apart the effects of locomotor strategy and size here. According to the craniofacial evolutionary allometry hypothesis (CREA) larger mammals tend to have smaller braincases and longer rostra. In your sample here, almost all of your fossorial species are large and most of your large species are fossorial. We don't think there's much you can do about this in terms of your sample, but it would be good to see an acknowledgement in the discussion that, although the patterns seen here may be related to locomotion, there are other possible explanatory factors.

This is a great suggestion. The regressions now include body mass as a Predictor variable either alone or as interacting with locomotion (Table 2). What we see is that not locomotion alone but the association between locomotion and body mass have an impact on brain size (Table 2). In other cases such as for the relative size of the petrosal lobules and neocortex, locomotion alone has an impact on the proportion of these brain regions.

Lines 26-29: The last two sentences of the abstract seem unrelated to the study here

Lines 15-26: We rewrote the abstract and deleted the last two sentences.

Line 33: Sciuridae is the third-most diverse rodent family (after Muridae and Cricetidae)

Line 56: We updated the text to Sciuridae being the third-most diverse rodent family.

Line 36: An explanation of the distinction between fossoriality and terrestrial mammals that burrow would be very helpful here. What distinguishes Aplodontia as fossorial compared to ground squirrels?

Lines 138-143: We used Samuels and Van Valkenburg (2008) to distinguish terrestrial rodents from fossorial forms. We have added a sentence in the method to explain our reasoning.

From Samuels and Van Valkenburg (2008):

Terrestrial are the Semifossorial from Samuels and Van Valkenburg (2008): “Regularly digs to build burrows for shelter, but does not forage underground (e.g., ground squirrels).”

Fossorial: “Regularly digs to build extensive burrows as shelter or for foraging underground (e.g., gophers and mole rats). Display a predominantly subterranean existence.”

Samuels JX, Van Valkenburgh B (2008) Skeletal indicators of locomotor adaptations in living and extinct rodents. *J Morphol* 269 (11):1387-1411. doi:10.1002/jmor.10662

Line 37: There seems to be growing agreement that the family name of mountain beavers should be spelt ‘Aplodontiidae’ (with two Is). It’s spelt this way in the Handbook of the Mammals of the World.

Line 60: We made this change throughout the text and figures.

Line 53: There is also considerable radiation of murids in Miocene which may have contributed to aplodontiid decline.

Lines 74-76: Although this is a very interesting point, we could not find a reference for the potential competition of murids and aplodontiids in North America. The competition between squirrels and aplodontiids was studied by Hopkins (2007) but she did not include murids in her study. In the absence of a study that has looked at this explicitly, we don’t feel we can mention the idea here, although we agree that it is an interesting hypothesis that merits future testing.

Hopkins SS (2007) Causes of lineage decline in the Aplodontidae: testing for the influence of physical and biological change. *Palaeogeogr, Palaeoclimatol, Palaeoecol* 246 (2-4):331-353

Lines 83-85: The numbers don’t add up here.

Lines 105-106: We updated the numbers.

Line 84: What was the resolution of the scans?

Line 107: We added a table summarizing the scanning parameters (Table S1).

Line 88: For some fossil species, you had multiple specimens from which you took a mean. Did you also try including them all as individuals in the dataset, to get an idea of the extent of intraspecific variation?

Lines 177-179: No, we did not look at intraspecific variation. We think it would be interesting and we could do it in the future. In particular, we have three specimens for *Ischyromys typus* and are working towards enhancing the sample for other taxa. Because there are few taxa in this sample that are represented by more than one individual, mixing in intraspecific variation for the small number that are represented by more than one individual would complicate the interpretations of the sample overall.

Line 109: Were the extant species assigned locomotor categories on the basis of ear morphology or from published behavioural studies?

Lines 137-138: The locomotor categories for extant taxa are based on behavioural studies. The sources are available in main references, Table S3 and in the supplementary file after the supplementary figures.

Line 114: This is a strong statement that underpins your analyses. It needs referencing.

Line 136: We added two references: Spoor et al. (2007) for the semicircular canals and Samuels and Van Valkenburg (2008) for the postcrania.

Spoor F, Garland T, Krovitz G, Ryan TM, Silcox MT, Walker A (2007) The primate semicircular canal system and locomotion. *Proceedings of the National Academy of Sciences* 104 (26):10808-10812. doi:10.1073/pnas.0704250104

Samuels JX, Van Valkenburgh B (2008) Skeletal indicators of locomotor adaptations in living and extinct rodents. *J Morphol* 269 (11):1387-1411. doi:10.1002/jmor.10662

Line 185: This list of variables is in a somewhat random order. Maybe list from largest to smallest?

Line 230: This section has been changed and does not include this sentence anymore.

Line 209 “For each relationship, we obtained two p-values, one for the slope (always zero) and one for the intercept, providing information about the significance of the relationship.” Do they really? Surely, the slope and intercept tell you something about

the relationship between the two variables, and it is the R-squared value that indicates the strength of the relationship. And why is the p-value of the slope always zero?

Lines 230-310: This section has been changed and does not include this sentence anymore.

Line 236: A permutation test is a better way (than Kruskal-Wallis) of assessing significance between groups in which the data isn't normally distributed

Lines 230-236: Great suggestion. We have updated this section and have included a permutation test instead of a Kruskal-Wallis test (Table 1).

Line 238: Why was phylogeny not considered here? Was the phylogenetic correction within PEQ considered sufficient to account for evolutionary history?

Lines 232-233: These boxplots represent the percentage of each region of the brain and the PEQ for each locomotor mode. We could not include phylogeny into these boxplots except for the PEQ as it is the only metric that is calculated using the phylogeny.

We are not aware of a post-hoc test that can include phylogeny on this type of data.

Line 248: 'simulation' should be 'simulations'

Line 328: The change was made.

Line 277: Tell us in which case the residuals were not normally distributed

Lines 333-422: This section has been changed and does not include this sentence anymore.

Line 292: The list of significant regressions with all its 'and' and 'vs.' can get very confusing to read. Is there a clearer way of presenting this? A similar thing happens at

Lines 333-422: This section has been changed and does not include this sentence anymore.

Line 294: The three consecutive adverbs in this sentence make for a head-spinning read. Could it be phrased a little more simply?

Lines 333-422: This section has been changed and does not include this sentence anymore.

Line 294: Are the petrosal lobule and neocortex larger relative to body mass or endocranial volume?

Lines 333-422: This section has been changed and does not include this sentence anymore.

Line 300: The boxplots seem like a key part of your results. It would be nice to see Figure S5 as part of the main manuscript rather than in the supplementary info.

Lines 230-236 and Figure 2: The boxplots are now part of the main manuscript.

Line 305: It would be very useful to have a version of Figure 2 with all the different locomotor types represented, so that the reader can see how they all compare to one another, not just the ones you went on to analyse further.

Lines 364-377 and Figure 3 (originally Fig. 2) have been updated and now include all locomotor types.

Figure 3. Phylogenetic corrected PGLS regressions of the **a.** Endocranial volume (cm³) vs. Body mass (g) ; **b.** Neocortical surface area (mm²) vs. Endocranial surface area (mm²) **c.** Olfactory bulbs (mm³) vs. Body mass (mg); **d.** Olfactory bulbs (mm³) vs. Endocranial volume (mm³); Petrosal lobule volume (mm³) vs. Body mass (mg); **e.** Petrosal lobule volume (mm³) vs. Endocranial volume (mm³). Data used for all regression are in Logarithm 10. **Abbreviations:** 1.EB, Endocranial volume ~ Body mass + Locomotion * Body mass; 1.PB, Petrosal lobule volume ~ Body mass + Locomotion*Body mass; 1.PE, Petrosal lobule volume ~ Endocranial volume + Locomotion; 2.NE, Neocortex surface area ~ Endocranial surface area + Locomotion; 2.OB, Olfactory bulb volume ~ Body mass + Locomotion; 2.OE, Olfactory bulb volume ~ Endocranial volume + Locomotion.

Line 323: To clarify, PEQ increase in Pteromyini was related to both a brain size decrease and body size decrease?

Lines 435-436: Yes, this is correct.

Line 333: 'surface area' rather than 'volume'

Line 441: We made the change.

Line 337: Do you mean that the neocortex expanded IN Sciuridae and Aplodontiidae, or should that be on the lineages leading to?

Lines 445-446: On the lineage leading to both of these clades.

Line 365: 'Sciurini' rather than 'Sciurine'

Line 471: We made the change.

Line 503: change 'to what is inferred' to 'to that inferred'

Lines 643-644: We made the change.

Line 507: Are there other fossil species out there that would help resolve this question? What would you liked to have scanned in an ideal world?

Line 647: We are not aware of any other fossil squirrels that could be CT scanned to help us answer this question apart from a flying squirrel which is not yet published, and so unavailable for inclusion in this analysis. It would be nice to have fossils belonging to other subfamilies (i.e., Xerinae, Callosciurinae, and Ratufinae)! But unfortunately, we are unaware of any relevant specimens.

Table S4: Minus sign missing from first number?

This table has been deleted.

Answers to reviewer #2:

Thank you for asking me to review the manuscript entitled “The impact of locomotion on the brain evolution of squirrels and close relatives during the Cenozoic”. This study incorporates several advanced phylogenetic approaches to present a thorough analysis of ~50 million years of brain evolution and ecological diversification among squirrels and relatives. It incorporates endocast metrics and phylogenetic data from both extant and extinct species and presents the results in visually pleasing figures. The study also introduces three entirely new endocasts from *Marmota marmota*, *Urocitellus richardsonii* and *Cynomys ludovicianus*, which are all nice additions. They have discerned significant ecomorphological trends in brain shape and estimated probabilities of potential ancestral states across various brain regions and niches. I enjoyed reading this manuscript and feel that it has much to offer. However, it still requires some substantial reformatting of the narrative and clearer justification for some of the methodological choices that have been made.

Thank you very much for this very useful feedback. We have greatly modified the Introduction so that it now sets the stage for why our study is important for broader questions about brain evolution (and not just rodent evolution), and we have clarified our objectives by ending the introduction with 3 key questions to be answered in the manuscript. We have also updated the method section concerning the role of locomotion on brain size and its components (see below for more details).

MAIN POINTS

1) A key problem with the arguments presented here is that the authors routinely conflate locomotion and overall ecology, the most obvious example being in the title itself. Locomotion strategies are even listed as “Ecology” in Table S1. While locomotion is undoubtedly a defining aspect of each niche represented by the taxa included, it is highly unlikely that locomotion alone is responsible for all their findings.

This is a good point. We do not mean to imply that locomotion is the only aspect of organismal ecology. Our focus is specifically on locomotion instead of other aspects of ecology. We have updated the text and replace “Ecology” by “Locomotion”. **Table S3** (previously Table S1) has been updated. In the introduction, we made sure that we considered locomotion as one aspect of ecology and acknowledged that others exist.

Interestingly, a large chunk of the discussion is committed to pointing out other selective pressures of different niches which may contribute to the findings of endocast shape, such as communication, vision, and audition, but there are many others, including foraging strategies, predator avoidance, and courtship behaviours, which are all likely more complex in arboreal environments than they are underground.

Lines 490-496 and lines 563-571: Yes, we agree with Reviewer 2 that other aspects of the biology of the animal might contribute to the changes that we observed. We have updated the discussion and including foraging strategies in the context of the petrosal

lobule size and predatory avoidance related to brain size. To our knowledge, no studies have specifically looked at the influence of courtship behaviour on the mammalian brain. These are things that we can hopefully address in future studies, but the biggest issue is that many of these aspects of ecology are not easily obtainable for fossils.

While the authors present a fairly sound case for the petrosal lobules having a clear association with locomotion, neocortical surface area is a more difficult sell. For example, in gliders the petrosal lobules are relatively smaller while the neocortex remains consistent with arboreal species (Figure 4). It seems to me this study is more associated with the effects of niche diversification on the brain in general, rather than locomotion alone. Teasing apart the contrasting conditions of each lifestyle would provide a more complete picture than what has been presented in the manuscript's current form.

Lines 524-542: We have updated our results and discussion and now it seems that all extant squirrels have an expanded neocortex. We restructured this part of the discussion to reflect the reviewer's comments. Very few studies have looked at the cortical architectural map of the neocortex of rodents. We have referred to the work of Krubitzer et al. (2011), who showed that, for example, arboreal squirrels have a bigger visual cortex while terrestrial squirrels have a bigger somatosensory and motor area. However, it might be impossible to know which area of the neocortex is expanded using virtual endocasts, and we acknowledge this fact in the discussion.

Krubitzer L, Campi KL, Cooke DF (2011) All rodents are not the same: a modern synthesis of cortical organization. *Brain Behav Evol* 78 (1):51-93. doi:10.1159/000327320

2) Abstract:

A little more detail on what kind of data was collected from the endocasts and what types of tests were carried out would help here.

We have greatly revised and added more details to the abstract. Unfortunately, we are limited by the space allocated for the abstract (150 words maximum).

3) Introduction:

There's some great points in this section. But it is short, at only three paragraphs, which undersells the significance of the work that has been put in and the results they present. There also seems to be a lack of clear direction. The paragraphs are a little disjointed and missing a clear rationale for conducting this study. The first paragraph details rodent/squirrel diversity, the second describes their evolutionary history. This information is of course important, but this is fundamentally a study on brain morphology and its association with ecological diversification through evolutionary time. Endocasts and their potential to resolve such associations are not mentioned until the third and final paragraph, and the methods used aren't specifically mentioned at all. More context should be brought to what is known about brain ecomorphology to date, and sooner in the Introduction. How is body size associated with brain size in these animals? And in

mammals in general? Is there modularity evident in brain allometry? What features of the brain are associated with the specific selective pressures of different niches? How do these features differ between environments? And across climates? Do these trends transcend any modularity of the brain associated with allometry? And are these brain/niche correlations consistent across mammalian taxa? Can a climatic influence on the evolution of brain shape therefore be discerned? How can endocasts be used to answer these questions? Most importantly, what is expected to be found from these analyses? And how could the findings be expected to improve our understanding of mammalian brain evolution in general? Some of these questions may be easy to answer with the current literature, some possibly less so. But as it stands, the authors have only briefly touched on them in a couple of sentences and have only provided two citations - both of which are their own previous work. There isn't a clear hypothesis presented, but rather the reason for the study seems to be simply because these tests haven't been done on these animals before. I believe a stronger argument can be made than that.

Lines 28-99: We really appreciate this suggestion, which is spot-on. Our original introduction was too short and did not do a good job of articulating the broader relevance of our study. We have restructured the introduction and made it longer and more detailed. It starts with a summary about what is known of the relationship between ecology and brain size (and its components), then transitions to why rodents are a great test case for studying how brain size and ecology and phylogeny are related over time, and then ends with three key questions that we will address in the manuscript, and then circle back to in our concluding paragraph. We hope that these changes have made our argument stronger.

Methods:

4) The authors have presented an extensive assortment of analyses, tables, and figures, both in the manuscript and supplementary. The first thing that confused me was the number of specimens used. There are 44 endocasts made from 38 species of rodent: 28 extant species and 13 extinct species. But $28 + 13$ is 41, not 38. And are some species therefore represented more than once to reach $n = 44$? But then the number of species presented in Table S1 is 38, so I'm not following. Then it is stated that the data set includes 14 endocasts of well-preserved crania of ischyromyids. Aren't these all extinct? If so, is it 13 extinct species used, as stated earlier, or is it 14? I trust that the authors know what's going on here, but I think a reduced version of Table S1 would be immensely helpful here, denoting the number of individuals of each species, their body mass, habitat/locomotion, and their extant/extinct status.

Lines 105-106: We have corrected the numbers in the text. We have added a new table: **Table S2**, which includes the subfamilies, species, locomotion, extant/extinct status, and collection number.

5) I have some concerns about measuring volume of the petrosal lobules from endocasts. Endocasts do not contain information on internal brain anatomy so it can be tricky to get reliable estimations of volume from some brain sections. While the olfactory bulbs are fairly easy to recognise as an isolated mass protruding from the brain, the petrosal lobules are less obvious in some species. So how is the functional volume of these structures below the surface morphology of the endocast reliably estimated?

Lines 172-176: We have added two sentences explaining that we used the anterior semicircular canal of the inner ear to isolate the petrosal lobules. In rodents, the function of the paraflocculi is entirely inside the subarcuate fossa, therefore we considered that the petrosal lobules accounted for the functional volume of these structures (McClure & Daron, 1971; Sakamoto et al., 2017).

McClure TD, Daron GH (1971) The relationship of the developing inner ear, subarcuate fossa and paraflocculus in the rat. *American Journal of Anatomy* 130 (2):235-249. doi:10.1002/aja.1001300209

Sakamoto A, Kuroda Y, Kanzaki S, Matsuo K (2017) Dissection of the Auditory Bulla in Postnatal Mice: Isolation of the Middle Ear Bones and Histological Analysis. *JoVE* (119):e55054. doi:doi:10.3791/55054

The authors even suggest in the Discussion that the observed smaller sizes of the petrosal lobules in the Pteromyini may be a result of them being obscured in some manner by the auditory bullae or distributed differently due to packing of the brain. If so, how can we know if they are in fact smaller or not? They may just be structured or positioned differently.

We deduced that the absence of the petrosal lobules was because the auditory bulla was very large and was therefore leaving no space for the subarcuate fossa to form. In other cases where the subarcuate fossa is visible, we could isolate the petrosal lobule and obtain its volume. The petrosal lobules are always in the same position compared to the rest of the cerebellum when these structures are present. So to the extent that we can tell, we are forming fair comparisons among the datasets.

6) The authors present two phylogenies for their sample, “Meng” and “Asher” and choose to present “Meng” in the manuscript and “Asher” in the supplementary. But no reasoning for this choice is provided. Was it arbitrary? They state that only minimal differences were found. If so, the supplementary figure probably isn’t required at all if the minor discrepancies are detailed in the manuscript for justification.

Lines 193-200: In the phylogeny from Meng (1990) the position of *Reithroparamys* is more in line with other work that has been done on this genus (Wible et al., 2005). Furthermore, *Reithroparamys* has an ossified auditory bulla and common canal for the facial nerve and the stapedial artery, which are present in squirrels but not in Paramyinae (Meng, 1990; Meng et al., 2003; Bertrand et al., 2019). For these reasons, we used the Meng (1990) phylogeny.

We have deleted the supplementary data that included Asher's results and only mentioned in the manuscript that the results using both phylogenies were similar.

Bertrand OC, Amador-Mughal F, Lang MM, Silcox MT (2019) New Virtual Endocasts of Eocene Ischyromyidae and Their Relevance in Evaluating Neurological Changes Occurring Through Time in Rodentia. *J Mamm Evol* 26 (3):345-371. doi:10.1007/s10914-017-9425-6

Meng J (1990) The auditory region of *Reithroparamys delicatissimus* (Mammalia, Rodentia) and its systematic implications. *American Museum Novitates* 2972:1-35

Meng J, Hu Y, Li C (2003) The osteology of *Rhombomylus* (Mammalia, Glires): implications for phylogeny and evolution of Glires. *Bulletin of the American Museum of Natural History* 2003 (275):1-247

Wible JR, Wang Y, Li C, Dawson MR (2005) Cranial anatomy and relationships of a new ctenodactyloid (Mammalia, Rodentia) from the early Eocene of Hubei Province, China. *Annals of the Carnegie Museum* 74 (2):91-151

7) I admit that I have limited experience with some of the more advanced phylogenetic regressions carried out. All tests make perfect sense to me though, at least up until the authors add each locomotor type to test their individual effect on the phylogenetic tests. The first part that concerns me is that each individual type of locomotion is tested for the most appropriate model via the Akaike's Information Criteria. I don't think I've ever seen all groups within a single discrete variable be individually tested for their own most appropriate model. Shouldn't each type of locomotion be standardised to be tested under the same model? Otherwise there may be a risk of over-manipulating the data, especially for such small sample sizes for each group. Are there other studies that have done this previously? I think, at minimum, the authors need to be clearer about what they are doing here and justify their decisions with citations of prior studies that employ similar methods when possible. The second part of this I'm unsure about is how sample size (N) can be consistent across each locomotion type, at N=37-38, when all locomotion groups are N≤10. The Terrestrial group only has an N of 6. So how is Terrestrial locomotion tested across all 38 species? Is it because each locomotion strategy is treated as a binomial presence/absence for each model? Wouldn't it make sense (and be easier) just to test locomotion as a categorical variable in the initial model and carry out pairwise comparisons between locomotion groups? I would like to see this section made a lot clearer.

Lines 230-310: Following this and other feedback from the other reviewers, we extensively modified our methods section, which we feel is now explained more clearly. We performed three different types of analyses: (1) permutation tests instead of Kruskal-Wallis tests. We have moved the boxplot in the main document; (2) PGLS analyses that considered locomotion as the predictor variable, rather than having a specific locomotor mode being the variable. We also included a figure illustrating how each locomotor mode differ from one another; (3) non-phylogenetic corrected ANOVA using Residual Randomization and associated post-hoc pairwise tests to look at how locomotor groups differ from one another. The reason for using non-phylogenetic post-

hoc tests stems from the fact that to our knowledge; it cannot currently be done following a PGLS analysis (Weisbecker et al., 2020).

The Asymptotic K-Sample Fisher-Pitman Permutation Test and pairwise comparisons were done on the PEQ, olfactory bulb, petrosal lobule, and neocortex percentages (Table 1). We selected 6 different regressions by using the AIC index (see Tables 2, 3). We made a summary of the PGLS process in Fig. S2. The ANOVA and non-phylogenetic corrected postdoc tests are in Tables S6, and 4. The code is also available in the supplementary data and in a GitHub repository.

Weisbecker V, Speck C, Baker AM (2020) A tail of evolution: evaluating body length, weight and locomotion as potential drivers of tail length scaling in Australian marsupial mammals. *Zool J Linn Soc* 188 (1):242-254. doi:10.1093/zoolin/znz055

8) It is a little strange that there is a paragraph detailing the generation of boxplots that are only presented in the supplementary. All methods in the main manuscript should be relevant to all results presented in the main manuscript. Many people would likely read how these boxplots were created and will never see them. The authors should probably consider whether to include the boxplots in the main manuscript or not at all.

Lines 333-345: The boxplots have been included in the main manuscript (**Fig. 2**).

Results and Discussion:

The Results section is great. Nice figures and findings that are easy to follow. The Discussion could use some restructuring to follow on from my earlier suggestions regarding the Introduction, so I'll leave that for now. But two things are worth a mention.

9) Firstly, the authors present temperature data and oxygen data in Figure 3, modified from Zachos et al. In the Discussion, they then mention the potential significance of climate in the evolution of these animals. However, there is no context provided prior to Figure 3 to indicate that climate was ever an important consideration of their study design. Climate and climate variables aren't mentioned to any degree of importance in the Introduction, nor mentioned at all in the Methods.

We did not statistically test for the effect of climate, as this would be exceedingly difficult for a dataset of our size and quality. We only generally looked at how brain size and its components changed through time with temperature, in a qualitative way. This is more of an indirect effect and we discuss this in the discussion. We found only one study (Benoit et al. 2019) that looked at the effect of temperature on brain evolution. In that, the effect of climate was not statistically tested either, but discussed to give some context to the observed changes. We feel that it would be very interesting to design a study to consider these factors specifically and quantitatively, but to do so well would require a sample design that was framed around that question.

Benoit J, Legendre LJ, Tabuce R, Obada T, Maraescul V, Manger P (2019) Brain evolution in Proboscidea (Mammalia, Afrotheria) across the Cenozoic. *Sci Rep* 9 (1):9323. doi:10.1038/s41598-019-45888-4

10) Secondly, a minor thing to address here is that, while these tests do demonstrate clear significant associations, they are still based on results that are limited by the diversity of the sample and variables tested. The authors need to make more careful use of language that acknowledges this point in both the results and discussion. Various events are sometimes presented as fact, rather than interpreting their results as supporting specific hypotheses. Results will always be limited to the data analysed (in this case, 38 out of more than 300 known species of squirrels and relatives), and so definitive statements should be kept to a minimum, particularly regarding reconstruction of ancestral states, which are based on probabilities generated from the taxa used. For example, the results SUPPORT the existence of an arboreal ancestor for Aplodontidae and Sciuridae. The results can't, and don't, simply demonstrate that their ancestor WAS arboreal. I'm sure the authors are aware of the difference and fully intend for the former interpretation to be received, but it's a minor linguistic detail that can lead to an incorrect understanding by readers.

Lines 423-474: We have made sure to apply these suggestions to the manuscript.

As I stated, I enjoyed reading this study. It represents a thorough analysis of the ecomorphology and evolution of the brain in a very charismatic group of mammals. I'm confident that the authors can address the aforementioned points and I hope that they are received in the constructive manner that is intended. I look forward to seeing the revision.

MINOR COMMENTS

11) Line 23. "We find that arboreal species have larger brains with bigger neocortices and petrosal lobules compared to fossorial taxa"
Absolutely or relatively?

Lines 15-26: The abstract has been rewritten and does not include this sentence anymore.

12) Line 25. "which may have facilitated the move into a complex three-dimensional environment"
Aren't all environments 3-dimensional? Suggest rewording. Dynamic perhaps?

Lines 15-26: The abstract has been rewritten and does not include this sentence anymore.

13) Lines 33-34. " Squirrels (Sciuridae), the second most diverse extant family of rodents..."
According Burgin et al., (2018) the Sciuridae are the third largest rodent family, behind Muridae and Cricetidae.

Line 56: We made this correction.

14) Line 37. Remove “hugely”.

Line 60: We made this correction.

15) Line 38. How diverse were the Aplodontidae?

Line 60: Hopkins (2008) used 98 known species for her phylogeny. We added this aspect in the text.

Hopkins SS (2008) Phylogeny and evolutionary history of the Aplodontoidea (Mammalia: Rodentia). Zool J Linn Soc 153 (4):769-838. doi:10.1111/j.1096-3642.2008.00399.x

16) Lines 38-40. “The striking differences between living sciurids and aplodontids raise the question of how the modern rodent assemblage emerged.”

This sentence hints that this study is addressing questions at the Rodentia scale. But it isn't. It's confined to sciurids and close relatives.

Lines 61-63: We changed the sentence.

17) Lines 41-42. “The likely common ancestors of squirrels and the mountain beaver, known as ischyromyids, were probably fossorial to scansorial.”

This is an odd way to start the paragraph, because there has been no prior context provided as to why locomotion is important in rodent evolution.

Lines 64-65: We changed the sentence.

18) Lines 50-51. “...Then, in the drier and more open environment of the Miocene (23-5.3 million years ago), aplodontids shifted to become predominantly specialized diggers, while squirrels continued to occupy the same Oligocene niches and added gliding to their repertoire.”

I'd be careful with this word usage. No niches are the “same”. Also, it's probably more correct to say that gliding evolved in some species, rather than the whole taxon “adding” gliding.

Lines 73-74: We changed the sentence.

19) Lines 65-66. “...the relationship between the form of the brain and ecology has yet to be assessed in these rodents over time, in a phylogenetic context, to test whether different locomotor groups had uniquely different brains and how these evolved.”

Is this the hypothesis? How are they expected to differ? What is expected to be teased apart from phylogenetic shape differences between modes of locomotion/ecology? What impact will the findings be expected to have on our understanding of mammalian brain evolution?

Lines 89-99: We updated this section and added 3 questions.

20) Lines 66-68. “Here we use a CT scan dataset of extinct and extant squirrels, apodontids, and close relatives, and assess changes in the brain over time, across phylogeny, and associated with ecology.”

How are you planning to do this? Any interesting and/or unique methods? These could be briefly mentioned here.

Lines 89-99: We updated this section, but we decide to not mention the methods because we wanted to keep the introduction short. Of course, we are happy to add methods in the introduction if the editor prefers.

21) Lines 75-80. Figure 1 looks nice. But most of it is not relevant, nor referenced in the text. Suggest instead to include just three angles of part C (dorsal, lateral, and ventral). Also, there is no ‘e’ in the Figure. All parts need to be labelled.

Figure 1 is referenced in the Material and Methods in section “Segmentation and endocranial quantification” (lines 147-181). We added the same colors on the dorsal and ventral views as in the lateral view. We prefer to keep part **a** and **b** because they show the specimen in its entirety and gives a better idea of how the endocast fits into the endocranial cavity. We decided to only label the regions that were under study.

Figure 1. Virtual endocast of the early Oligocene squirrel *Cedromus wilsoni* (USNM 256584) based on computed tomography (CT) data **a.** lateral view inside a solid cranium; **b.** lateral view inside a translucent cranium; **c.** lateral view with the different regions estimated in the paper highlighted; **d.** dorsal view; **e.** ventral view. **Abbreviations:** **nx**, neocortex; **ob**, olfactory bulb; **pl**, petrosal lobule. Scale bars equal 10 mm.

22) Line 108. A Table would help so much here. See previous comments.

Table S2 was added (see above).

23) Line 114. Citation needed after "...which are well validated proxies for locomotor behaviour."

Line 137: We added two references: Spoor et al. (2007) for the semicircular canals and Samuels and Van Valkenburg (2008) for the postcrania.

Spoor F, Garland T, Krovitz G, Ryan TM, Silcox MT, Walker A (2007) The primate semicircular canal system and locomotion. *Proceedings of the National Academy of Sciences* 104 (26):10808-10812. doi:10.1073/pnas.0704250104

Samuels JX, Van Valkenburgh B (2008) Skeletal indicators of locomotor adaptations in living and extinct rodents. *J Morphol* 269 (11):1387-1411. doi:10.1002/jmor.10662

24) Lines 185-187. "For each taxon we obtained six variables: endocranial volume, body mass, olfactory bulb volume, petrosal lobule volume, neocortical surface area and endocranial surface area."

Some of these have already been mentioned. Suggest moving this passage up to before any methods for measuring them have been described.

Lines 230-310: This section has been changed and does not include this sentence anymore.

25) Line 262. Table S5 is first mentioned after Table S6. Their names should be reversed.

Lines 299-310: This section has been changed and does not include this sentence anymore.

26) Lines 265-268. I'm not following everything going on in Table 1. Each of the p-values need to have a separate heading.

This table (previously Table 1) has been deleted.

27) Lines 272-272. "The models with the lowest AICc values were chosen as the best-fit model for each analysis (Tables S4, S7)."

This sentence is not needed here. Should be mentioned in the methods already.

Lines 333-377: This section has been changed and does not include this sentence anymore.

28) Lines 273-275. "Phylogenetic generalized linear squares models (PGLS) always provide a better fit to our data than the linear squares regression (OLS), except for one regression using Meng's topology (see Table 1)".

OLS is mentioned in Table 1 but PGLS isn't. The Table should have both PGLS and OLS defined to avoid confusion from this sentence.

Lines 333-377: This section has been changed and does not include this sentence anymore.

29) Lines 306-310. Have these regression lines been generated for each group independently? I find it hard to believe they would all be perfectly parallel – particularly Fossorial in part D.

Lines 333-377 and Figure 3 (originally Fig. 2) have been updated and now includes all locomotor types.

Figure 3. Phylogenetic corrected PGLS regressions of the **a.** Endocranial volume (cm³) vs. Body mass (g) ; **b.** Neocortical surface area (mm²) vs. Endocranial surface area (mm²) **c.** Olfactory bulbs (mm³) vs. Body mass (mg); **d.** Olfactory bulbs (mm³) vs. Endocranial volume (mm³); Petrosal lobule volume (mm³) vs. Body mass (mg); **e.** Petrosal lobule volume (mm³) vs. Endocranial volume (mm³). Data used for all regression are in Logarithm 10. **Abbreviations:** 1.EB, Endocranial volume ~ Body mass + Locomotion * Body mass; 1.PB, Petrosal lobule volume ~ Body mass + Locomotion*Body mass; 1.PE, Petrosal lobule volume ~ Endocranial volume + Locomotion; 2.NE, Neocortex surface area ~ Endocranial surface area + Locomotion; 2.OB, Olfactory bulb volume ~ Body mass + Locomotion; 2.OE, Olfactory bulb volume ~ Endocranial volume + Locomotion.

30) Lines 359-360. “The locomotor behavioral ancestral state reconstruction showed that the common ancestor of rodents was scansorial (Fig. 5).”
MOST LIKELY scansorial, based on your results.

Lines 464-465: The change was made.

31) Lines 379-381. “The PGLS and OLS regressions show that arboreal species have significantly larger brains, petrosal lobules and neocortices compared to other locomotor modes...”
Relatively or absolutely?

Lines 477-601: This section has been changed and does not include this sentence anymore.

32) Line 422. I don't think the research cited here states this correlation.

Lines 477-601: This section has been changed and does not include this sentence anymore.

33) Lines 432-434. “The ancestral state reconstruction of the Marmotini tribe (ground squirrels) shows a small decrease in the proportion of the petrosal lobules and neocortex...”

I can see it in the petrosal lobules and PEQ but not the neocortical surface area in Figure 4. Am I missing something?

Lines 477-601: This section has been rewritten. These ancestral states are not indicated in Fig. 4 for lack of space. However, all the data about changes from one node to another are available in **Table S5**. Marmotini is node 28 and the previous node is 18 (see **Fig. S2**).

34) Lines 467-469. “Our new study reveals a more complex pattern involving expansions and reductions of different parts of the brain.”
Good opportunity to discuss evidence for modularity here.

Lines 607-609: To our knowledge, there are no studies that have looked at modularity and brain shape/size in mammals. We only found a paper talking about cranial modularity and connected it to brain size in mammals, but the modularity of brain regions is not discussed. Therefore, we are not sure how to discuss links between modularity and brain size without proper testing using 3D geometric morphometrics to look at how the shape of brain regions vary and if they belong to specific modules. This would be an interesting topic for future study, and we will strongly consider it.

Koyabu D, Werneburg I, Morimoto N, Zollikofer CPE, Forasiepi AM, Endo H, Kimura J, Ohdachi SD, Truong Son N, Sánchez-Villagra MR (2014) Mammalian skull heterochrony reveals modular evolution and a link between cranial development and brain size. *Nature Communications* 5 (1):3625.
doi:10.1038/ncomms4625

Answers to reviewer #3:

The present manuscript evaluates the role of locomotion (scansorial, fossorial, gliding etc.) in shaping the evolution of the brain in rodents (squirrels) over several millions of years. The authors use CT scans and virtual endocasts of the braincase to reconstruct the size of portions of the brain in fossil and extant squirrels. They then analyze the relationships among locomotion, body mass, and the size of various regions of the brain as well as use ancestral character estimation to reconstruct the likely ancestral states of brain characteristics and body size. The authors also form some hypotheses relating changes in global climate during the period of squirrel evolution to the appearance of new locomotor types and diversification within those groups. This is the first study I am aware of to combine fossil and extant squirrel species into an analysis of brain size. The analysis also covers a very large portion of the Cenozoic, also making it the most temporally expansive study of its kind, to my knowledge. In this sense, the paper is quite fantastic.

I am primarily concerned about the statistical analyses for two reasons 1) a lack of clarity regarding how the models were actually fit, 2) I don't believe the authors are really testing the hypotheses they think they are (see below).

Thank you very much for this very useful feedback. We have updated the analyses (see below)

My comments on the methods and results will reflect a general confusion I had that stemmed from a lack of clarity in the text. I only figured out (maybe?) how the models were fit after looking at Figure 2. This is insufficient. The text should say "we performed these univariate regressions for all taxa and then for subsets of taxa within each locomotor group."

We have updated the analyses (see below)

But then this brings me to my second concern regarding the hypotheses they are testing. Throughout the discussion, in particular, the authors state things like "locomotor mode had a significant impact on relative brain size etc." But most of the analyses they use do not actually test this (in fact, only the pairwise comparisons of box plots does). What they actually test with the regression analyses is whether the relationships between brain morphology and body mass etc. vary among groups with different locomotor modes. Based on the questions set up in the manuscript, I can't understand why the authors would opt for this approach over a multiple regression approach that simultaneously tests for differences in brain morphology among groups and say brain morphology and body mass.

Lines 95-99: We now do a series of multiple regressions, following the reviewer's suggestion.

Perhaps, my confusion simply stems from the fact that their hypotheses are not clearly communicated at the outset of the paper. In which case, this is an easy fix with little to no re-analysis. But based on the way things are worded in the discussion, I am uncertain.

We now describe our tests with more clarity, and have added 3 questions to the introduction that frame all of the statistical analyses we are doing, which sets out our major goals from the outset.

Introduction

Paragraph starting on line 57 – The authors don't present any citations for their interpretations of the functions of the different brain regions. I think this is an oversight (and, in fact, the citations I was looking for show up in the discussion. They should probably also be cited in the introduction).

Lines 28-99: We updated the introduction, and it includes more citations now.

The introduction does not present any particular hypotheses. This may really have helped later on when I was trying to discern why the authors fit the models they did. It might also help parse out to what degree the paper is focused on correlating brain morphology and ecology and change in brain morphology through time. It seems to be both but the introduction almost makes it seem like the paper will be more focused on the time component.

Lines 28-99: We have restructured the introduction and made it longer and more detailed. It starts with a summary about what is known of the relationship between ecology and brain size (and its components), then transitions to why rodents are a great test case for studying how brain size and ecology and phylogeny are related over time, and then ends with three 3 key questions that we will address in the manuscript, and then circle back to in our concluding paragraph.

Methods

Although the sample size is small relative to the diversity of rodents, I acknowledge the time and resource intensive nature of CT scanning and creating the virtual endocasts used here.

BUT I do wonder about the statistical power of their analyses. There are very few samples per locomotor group, and so the power to detect differences (or to perform regressions on subsets of the data) may be particularly low (though, the fact that they do detect differences probably means that more samples would only strengthen the relationship).

We take the reviewer's point. Although we have not been able to sample the full diversity of extinct and extant squirrels and close relatives, our sample is considerably larger than any that has been compiled previously. We also feel that it is a representative sample. We have added four new terrestrial taxa, which greatly increase the taxonomic, body mass and locomotor range of our sample compared to previous studies. We agree with the reviewer: increased sampling in the future should only strengthen our main findings, and add further statistical significance to the significant relationships we are already able to find with our current dataset. We will endeavour in the future to continue to CT scan fossil and modern specimens to expand our sample for follow-on studies.

Line 114 – states that there is a validated relationship between inner ear/post-cranial morphology and locomotor behavior but no citations are provided.

Line 136: We added two references: Spoor et al. (2007) for the semicircular canals and Samuels and Van Valkenburg (2008) for the postcrania.

Spoor F, Garland T, Krovitz G, Ryan TM, Silcox MT, Walker A (2007) The primate semicircular canal system and locomotion. *Proceedings of the National Academy of Sciences* 104 (26):10808-10812. doi:10.1073/pnas.0704250104

Samuels JX, Van Valkenburg B (2008) Skeletal indicators of locomotor adaptations in living and extinct rodents. *J Morphol* 269 (11):1387-1411. doi:10.1002/jmor.10662

Line 119 – no citation provided to support using cheek tooth area as a predictor of body mass

Line 146: This is the same citation as the cranial length as a predictor of body mass. The citation is now next to both sentences.

Bertrand OC, Schillaci MA, Silcox MT (2016) Cranial dimensions as estimators of body mass and locomotor habits in extant and fossil rodents. *J Vert Paleontol* 36 (1). doi:10.1080/02724634.2015.1014905

The method of producing phylogenetic topologies and time calibrating them is sound, to my knowledge.

Phylogenetic regressions – This is where I start to get a bit confused about how the authors set up their regression analyses. I think this stems from the difference in how I understood it was done from the text, how Table 1 is structured, and how Figure 2 is presented.

From the text, I assumed that univariate models were fit (e.g. endocranial volume vs body mass) and then multivariate models were fit (e.g. endocranial volume vs. body mass and locomotor mode).

However, table 1 suggests that the authors only employed regressions with one independent and one dependent variable. What is the hypothesis being tested here? From the introduction, I thought the main goal of the manuscript was to test for a relationship between ecology and brain morphology/body mass. So, univariate regressions don't seem like the right approach to me.

I would think they authors would need to use multiple regression. For example:

Endocranial volume ~ body mass + locomotor behavior

Or

Endocranial volume ~ body mass + locomotor behavior + body mass:locomotor behavior (with this representing the interaction between the two variables)

These would simultaneously test for the relative effects of body mass and locomotor behavior (as well as any interaction between the two because, for example, arboreal rodents don't tend to be as large as marmots).

But things get more confusing on Line 191, where the authors suggest that they did add locomotor behavior to their regressions. But coming back from looking at Figure 2, it seems that the authors performed regressions for all of the species and then just for subsets of species from particular locomotor modes. How are these regressions relevant to the questions set forth in the study? Wasn't the goal to look for a relationship between ecology and the brain? These test whether the relationship between the brain/body mass etc. are different among groups with different ecologies. A different question entirely.

The authors could have simply done step wise model fitting, picked the best model based on AIC, and run with that rather than first fitting all of the univariate combinations.

Lines 230-310: The reviewer is absolutely correct, and these are very helpful suggestions. We now follow the reviewer's advice and use a series of multiple regressions, which are selected based on step wise model fitting. In general, as outlined above, we extensively modified our methods section. We now perform three different types of analyses: (1) permutation tests instead of Kruskal-Wallis tests. We have moved the boxplot in the main document; (2) PGLS analyses that considered locomotion as the predictor variable, rather than having a specific locomotor mode being the variable. We also included a figure illustrating how each locomotor mode differ from one another; (3) non-phylogenetic corrected ANOVA using Residual Randomization and associated post-hoc pairwise tests to look at how locomotor groups differ from one another. The reason for using non-phylogenetic post-hoc tests stems from the fact that to our knowledge; it cannot currently be done following a PGLS analysis (Weisbecker et al., 2020).

The Asymptotic K-Sample Fisher-Pitman Permutation Test and pairwise comparisons were done on the PEQ, olfactory bulb, petrosal lobule, and neocortex percentages (Table 1). We selected 6 different regressions by using the AIC index (see Tables 2, 3). We made a summary of the PGLS process in Fig. S2. The ANOVA and non-phylogenetic corrected postdoc tests are in Tables S6, and 4. The code is also available in the supplementary data and in a GitHub repository.

Weisbecker V, Speck C, Baker AM (2020) A tail of evolution: evaluating body length, weight and locomotion as potential drivers of tail length scaling in Australian marsupial mammals. *Zool J Linn Soc* 188 (1):242-254. doi:10.1093/zoolin/znz055

Line 193 – keep in mind that the strength of a relationship is only indicated by r^2 or other effect sizes and not p-values. The methods and hypotheses need serious clarification. The way the models were fit is exceedingly unclear until you get to Figure 2, then it was like an “AH HA” moment.

We updated our analyses (see above).

Line 199 – the authors mention they compare the fit of models assuming different correlation structures among the variables but say nothing about which variables actually went into these models.

We updated our analyses (see above).

This also seems like a very non-traditional approach to selecting between phylogenetic and non-phylogenetic regressions. Typically, one would test for phylogenetic signal in the residuals for the OLS and then implement the phylogenetic regressions. That being said, I don't think the approach is necessarily wrong.

Lines 254-265: We changed the original way of selecting for the model. First, we ran the OLS and checked how the residuals were organized with “phylogeny”. If closely related species showed a similar pattern, we used PGLS instead (see supplementary code for residual plots).

Table 1 – The caption needs to be clarified. In combination with the lack of clarity in the text, I did not understand why the table had 6 parts until my “eureka moment” with Figure 2.

Original Table 1 has been deleted.

Line 284 – The authors use very non-standard statistical terminology here. “Three lifestyles have an impact on specific relationships.” What does this mean? I now deduce that they mean that the relationship between brain/body mass differs among species from the different locomotor groups. Please simply say that. The way it is stated now is simply incorrect and misleading.

Lines 333-377: This section has been changed and does not include this sentence anymore.

In general, all of the wording with “when locomotor mode was included” is misleading. They did not include locomotor mode in the analyses. They analyzed subsets of the data for relationships among brain components/body mass.

We updated our analyses (see above).

Line 294 – why do these analyses come after the regressions? They seem to set up some basic information that a reader might want to know before diving into the model results.

Lines 333-345 and Figure 2: The boxplots are now part of the main manuscript and have been moved before the PGLS regressions.

Figure 2 – The caption needs to describe why they were two regression lines. I now understand that one is the regression of all species? And the other is just the subset from a particular locomotor group? Please make this clearer. The power of some of these individual locomotor mode regressions must be extremely low.

Figure 3 (originally Fig. 2) and the analyses have been updated (see above).

Figure 3. Phylogenetic corrected PGLS regressions of the **a.** Endocranial volume (cm^3) vs. Body mass (g) ; **b.** Neocortical surface area (mm^2) vs. Endocranial surface area (mm^2) **c.** Olfactory bulbs (mm^3) vs. Body mass (mg); **d.** Olfactory bulbs (mm^3) vs. Endocranial volume (mm^3); Petrosal lobule volume (mm^3) vs. Body mass (mg); **e.** Petrosal lobule volume (mm^3) vs. Endocranial volume (mm^3). Data used for all regression are in Logarithm 10. **Abbreviations:** 1.EB, Endocranial volume ~ Body mass + Locomotion * Body mass; 1.PB, Petrosal lobule volume ~ Body mass + Locomotion*Body mass; 1.PE, Petrosal lobule volume ~ Endocranial volume + Locomotion; 2.NE, Neocortex surface area ~ Endocranial surface area + Locomotion; 2.OB, Olfactory bulb volume ~ Body mass + Locomotion; 2.OE, Olfactory bulb volume ~ Endocranial volume + Locomotion.

Figure 3 – the raw data are available as part of the Zachos et al. (2008) paper.

We updated **figure 4** (originally Fig. 3) with the raw data from Zachos et al. (2008).

Figure 4. Ancestral state reconstruction of the Phylogenetic Encephalization Quotient (PEQ) with a geological time scale. The temperature and oxygen level ($\delta^{18}O$) plot during the Cenozoic is modified from Zachos et al.⁴⁰.

Figure 3 and 4 are really nice (except the Zachos curve).
Why not put the cute animals on Figure 3 and 4 as well?

For space reasons, if we add the images on the side, it will reduce the space for the important parts in **Figs. 4 and 5** (previously Figs. 3 and 4), which will make the figures difficult to read at publication size. However, we can add them if the editor would like us to.

Line 379 – an impact on the relationship between brain morphology and body mass etc. I think the way this is worded is throwing me off. It also uses the word significant in a way that I don't think it should be used. The authors aren't presenting a regression of locomotor mode on these other variables.

Line 477-489: We changed the sentence.

Line 382 – but you don't test this with the models.

Lines 490-601: We updated this paragraph and changed this sentence.

388 – a lot of the citations are was searching for in the introduction are here.

Line 410 – but your models don't test this (only the boxplots do)

Lines 490-601: We have updated our analyses, this section has been changed.

– significant impact, again this wording seems inappropriate

Lines 490-601: This section has been changed and does not include this sentence anymore.

Reviewers' comments:

Reviewer #1 (Remarks to the Author):

The authors have clearly spent a lot of time revising this manuscript and it is much improved as a result. The major concerns that we had in our original review have now been addressed i.e. the statistical analyses. The use of permutation tests and PGLS models has much greater validity, I think, than the analyses in the original manuscripts, and thus the conclusions that are drawn are much more robust. The more sophisticated models used here also address the concerns that we originally had about teasing apart the influences of body mass and locomotion on the brain.

I just have a few more minor issues that need clearing up, listed below. Otherwise I am satisfied that the authors have addressed all my concerns.

Line 139: Van Valkenburgh misspelt (final H missing)

Line 230: From Table 1, it seems you have used some kind of multiple comparisons adjustment. This needs to be described here (which method of correction did you use?).

Line 259: Would be good to list all the equations tested in a supplementary table alongside their AIC values.

Lines 277-289: I think you've made an error in the abbreviations here – shouldn't the final one be 2OE (i.e. set 2, Olfactory bulb, Endocranial volume)? See also comment on Tables 2-3 below.

Line 558: 'lead' should be 'led' or 'leads'

Line 564: There seems to be a word missing after 'terrestrial' – the other two terrestrial what?

Line 600: Probably worth noting here that many mammalian species lack the petrosal lobule / subarcuate fossa (including some other rodents and ourselves), but presumably still retain the functional elements of those structures since they are able to stabilize their gaze. See Gannon et al (1988, Am J Phys Anthropol, 77: 143-164) for a list of species lacking the subarcuate fossa.

Tables 2-3: I think there's mistake in the abbreviations (although a different one from above). Shouldn't 2OB, be 1OB? All the equations that use body mass as a predictor should be set 1, right? That would make the tables less confusing because the regressions would be presented in a more logical order i.e. 1EB, 1PB, 1OB, 2NE, 2PE, 2OE.

Philip Cox
University of York

Reviewer #2 (Remarks to the Author):

Thank you again for the opportunity to review the revised manuscript entitled "The impact of locomotion on the brain evolution of squirrels and close relatives". I appreciate the authors addressing all of my comments. The work they have done on the manuscript is commendable and their clarifications regarding my concerns appear logical and reasonable. The manuscript is well-researched and the study is supported with a strong rationale. I have just a few additional points to bring up.

MAIN POINTS

The authors have done a great job revising their tests. And for the most part, these are well done. However, there are some choices that could perhaps benefit with some further consideration. I have listed these below with some other brief notes.

Abstract:

Suggest including (1) how many species were used, and (2) the use of digital endocasts.

The word count can be reduced by removing word clusters like “and three components”, “we find that”, etc.

Introduction:

This is very much improved.

Methods:

The permutational tests make little sense to me from the descriptions from line 230. There are no other studies cited in this section that provide grounds for using them, so I’m not following their purpose. Why are these tests better than running the same regressions as for later tests, but on Locomotion as a single predictor? A concern regarding Fisher-Pitman Permutation tests is that in order to compare group means, this test requires homogeneity of variances across groups (Boik, 1987). This is because Fisher-Pitman Permutation tests can identify any types of differences between group distributions and so it may not necessarily demonstrate a difference between means if there is a significant difference between the variances (Neuhäuser & Manly, 2004). Some of the distributions in Figure 2 suggest that variances between some groups are not homogeneous. But I suppose variances can be compared with an F-test or something similar to verify the conditions.

Boik, R. J. (1987). The Fisher-Pitman permutation test: A non-robust alternative to the normal theory F test when variances are heterogeneous. *British Journal of Mathematical and Statistical Psychology*, 40(1), 26–42.

Neuhäuser, M., & Manly, B. F. (2004). The Fisher-Pitman permutation test when testing for differences in mean and variance. *Psychological reports*, 94(1), 189-194.

In the paragraph of lines 254-265, the authors state that they choose not to incorporate phylogenetic relations to test all models; their reason being that “in some instance OLS outperforms phylogenetic generalized least squares (PGLS)”. They assessed whether a PGLS analysis is appropriate for each model by plotting the residuals, followed by what seems to be a visual assessment of the residual distributions. While an OLS may perform better sometimes, I would still argue that all tests should be consistent for each model. Firstly, they are different tests, so the results for each aren’t strictly comparable. Also, a subjective decision based on visual inspection may miss something that is not immediately apparent in the eye-balling process - even mild trends that are hard to see can sometimes be statistically relevant. Finally, if no phylogenetical signal is evident, then relatedness won’t significantly impact the outputs anyway. I think it would be more thorough and reliable to either run both OLS and PGLS or just PGLS alone on all models, rather than choosing one or the other.

The authors cite Weisbecker et al. for not performing phylogenetic pairwise comparisons (line 302). Which is fine. But if the authors are really interested in pairwise comparisons on phylogenetically adjusted distributions, this may be possible using the ‘*phylANOVA*’ function in the *phytools* package. This function performs a simulation-based phylogenetic ANOVA and can then carry out posthoc comparisons of means among groups. However, this function only permits a single predictor variable containing groups. It could be used with Locomotion alone, but with additional variables is tricky. Body size could possibly be included as an initial predictor by using the residuals of the body size/measurement (e.g. petrosal lobules) correlation as the response variable and Locomotion as the predictor. Just some food for thought.

There are also some sections of the methods that could be streamlined. This is more an editing issue, so I won’t list things at length. An example, however, would be combining lines 239-243 and lines 243-247. These are both lengthy sentences that essentially provide the same information.

Results

These are fine, based on the tests performed.

Discussion/Conclusion:

These are also nice. The conclusion rounds everything up very well.

MINOR COMMENTS

Line 229: "Additionally, we also obtained the residuals from the PGLS regression endocranial volume vs body mass."

--This feels like half a sentence to me. Why was this mentioned? What was done with the residuals once they were obtained?

Line 232: "Phylogeny was only considered here in the calculation of the PEQ."

--Should mention why/why not for the others

Lines 279-286: The regressions in the methods are listed as 1EB, 1PB, 1OB, 2NE, 2PE, 2OB. But in the Tables as 1EB, 1PB, 1PE, 2NE, 2OB, 2OE. I think these models do not correspond. Some are olfactory bulbs while others are petrosal lobules.

Line 349: "p-value= 0.000"

--Remove '-value' here and elsewhere throughout the results (except table captions for defining "p"), and instead write as 'p < 0.001' in this instance.

Line 556-559: "Knowing the role of the petrosal lobules in maintaining the position of the eyes when the head rotates (vestibulo-ocular reflex) and in the tracking of moving object (smooth pursuit) lead us to hypothesize that these functions may be crucial when living in the 3D complex environment of the trees.

--This sentence could be more impactful in the Introduction. Perhaps somewhere in paragraph 80-88, which would lead to the question regarding their relative sizes across species/Locomotion groups.

Line 1029: Figure 1 is still missing an 'e' in the image for part e.

I would also encourage referring to this figure in the introduction so that the reader can see what an endocast is and get their bearings on the different parts.

Line 1042: I think there is a lot of repeated information here between the figure and the caption. Perhaps replace the section definitions with the abbreviations since the definitions are also the axis labels.

Line 1076: I'm not sure this table is necessary. The "model" column can be incorporated into Table 3 and the rest is detailed in the caption for Table 3 anyway.

Line 1079: PGLS, not PLGS. And suggest rewording the title. What's "different" about this analysis?

Reviewer #3 (Remarks to the Author):

Please forgive me for my delay in returning this review, especially given how short my re-review has turned out to be. 2021 continues to be as weird as 2020.

I reviewed a previous iteration of this manuscript. My primary concerns related to the lack of clarity regarding the statistical analyses and whether the analyses the authors used were, in fact, most appropriate. The manuscript is very much improved in clarity and I believe that it should be accepted for publication. It is an excellent and interesting work.

The questions are now clearly laid out in the introduction. This greatly enhanced my understanding of the goals of the study.

The authors have applied a new series of appropriate models for the questions posed in the manuscript and the methods are quite clear. It appears to address all of my original concerns.

I have made a very small number of specific comments on the PDF. Take them or leave them.

Thank you for the excellent read.

**The impact of locomotion on the brain evolution of squirrels and close relatives**

Ornella C. Bertrand¹, Hans P. Püschel¹, Julia A. Schwab¹, Mary T. Silcox² and Stephen

5 L. Brusatte¹

¹School of GeoSciences, University of Edinburgh, Grant Institute, Edinburgh, Scotland,

UK

²Dept. of Anthropology, University of Toronto Scarborough, 1265 Military Trail

Scarborough, Toronto, ON, Canada M1C 1A4

*Corresponding author: ornella.bertrand@ed.ac.uk

[revised manuscript text omitted]

is also an increase in the absolute size of the neocortex and petrosal lobules. In
contrast, the aplodontiid node shows a decrease in brain size and body mass,
associated with a slight decrease in PEQ, and a decrease in the absolute size of the
neocortex and petrosal lobules. Overall, the brain of early arboreal aplodontiids is
downscaled compared to early arboreal squirrels. This suggests that very early in the
evolution of Sciuroidea, squirrels and aplodontiids started manifesting different brain
organization trajectories.

In later fossorial Aplodontiidae, a shift occurred with a decrease in the proportion
of the neocortex and the petrosal lobules, which, as discussed above, might reflect the
less important role of these structures in animals spending more time underground. The
higher PEQ at this point in the phylogeny is due to a greater increase in brain size
compared to body mass. Thus, the brain was upscaled with an increase in the absolute
size of the endocast and neocortex, while the petrosal lobules became even smaller.
The change in brain size might have been a response to the body mass increase,
because the neocortex proportion is lower at this node.

At the node leading to extant subfamilies of squirrels, the PEQ and the proportion
of the neocortex and petrosal lobules continued to increase. The oldest known fully
arboreal member of Sciurini, the extinct *Protosciurus*, had a very similar brain to that
inferred for its immediately ancestral node. In contrast, modern-day Sciurini have a
much higher PEQ, with the neocortex comprising a greater proportion of brain size. This
means that after the Miocene, the brain of these arboreal species continued to become
larger and more complex. No other fossil crania are currently available to CT scan for

other still-surviving squirrel tribes, so it remains unclear if similar increases in
neocortical proportions occurred before or after the Miocene in these extant groups.

Climate and temperature may have indirectly affected, or ultimately driven,
neurobiological changes in rodent brains and sensory systems over the Cenozoic.
During the humid, hot, tropical climate of the early to middle Eocene, scansorial and
fossorial Ischyromyidae were diverse on the northern continents^{36,37,74}, perhaps as a
result of competition with primates, who had already seized many arboreal niches¹²³.
North American ischyromyids and primates started to decline in species diversity during
the global cooling event of the late Eocene^{38,40,123,124}, which is when arboreal sciuroids
appeared in the fossil record^{33,39}. This environmental change may have indirectly
enabled aplodontiids and squirrels to invade arboreal niches once held by primates,
which by then were less diverse. Arboreal animals generally have a lower body mass
compared to terrestrial species¹²⁵ and the noticeable decrease in body mass occurring
in Sciuroidea at this time, may have triggered their transition into the trees. Because
these body size decreases were not matched by concordant brain size decreases, PEQ
increased, and regions of the brain crucial for living in a 3D environment continued to
expand, with the larger neocortex indicating better vision, motor skills and/or
memorization, and larger petrosal lobules suggesting improved head and eye
movement control. In contrast, while many sciuroids were colonizing the canopy, the
Oligocene fossorial *Ischyromys* retained a relatively smaller neocortex and petrosal
lobules.

The less dense forests and more open woodlands of the Oligocene on the
northern continents, which probably played a role in the continued decline of primate

and ischyromyid communities, were the stage on which squirrels and aplodontiids
diversified. These more open environments were maintained during the Miocene and
may have promoted the evolution of highly specialist aplodontiid burrowers⁴³, with their
smaller brains, neocortices, and petrosal lobules. Ground squirrels were also diverse
during the Oligocene and Miocene and survived past the end of the Miocene while
aplodontiids waned⁴³ during a time when grasslands spread even further as climates
became cooler and drier. Ground squirrels may have been better adapted to grasslands
due to the larger neocortex they inherited from their more immediate tree-dwelling
ancestors. Among their many utilities, large neocortices have been linked to sociality in
primates¹²⁶, suggesting that ground squirrels may have maintained a large neocortex for
a different purpose than living in the 3D arboreal environment of their ancestors.

**Conclusions**

Returning to the key questions we posed at the outset of our study, we found that
(1) body mass is highly correlated with the relative size of the brain and its components,
while phylogeny impacts overall brain and neocortical sizes the most. Locomotion
correlates with the relative size of the brain, petrosal lobule, and neocortex, as well as
the absolute petrosal lobule size. For (2), arboreal and gliding species have larger
brains than others, particularly fossorial species. We also found that arboreal and
scansorial taxa have relatively larger petrosal lobules compared to fossorial taxa.
Furthermore, extinct arboreal rodents have a greater neocortical size compared to
extinct scansorial and fossorial species. For (3), we showed that arboreality and
fossoriality impacted the evolution of the brain in Sciuroidea. Squirrels and aplodontiids

show strikingly distinct ecomorphological diversities today and locomotion appears to
have dramatically shaped the evolutionary history of these two groups as they adapted
to new environments. This process took millions of years as the environment shifted
from early Eocene tropical forests to expansive Miocene grasslands. The transition from
scansorial Ischyromyidae to arboreal Sciuroidea included an increase in relative brain
size. Our finding that the neocortex and petrosal lobules expanded before the
Sciuroidea node, and continued to enlarge afterwards, highlights the potentially crucial
role of these brain components in the process of becoming arboreal. Squirrels and their
closest relatives demonstrate the connection between locomotion and brain evolution
and help us better understand how this group of rodents evolved in concert with their
environment over millions of years, to produce today's astounding diversity.

**Data availability statement.** The new surface rendering of the endocast of
*Reithroparamys delicatissimus* and those of the three extant terrestrial squirrels
generated for this project as well as the surface renderings for all the other specimens,
are available in MorphoSource (www.morphosource.org¹²⁷) at
https://www.morphosource.org/Detail/ProjectDetail/Show/project_id/83.

**Code availability.** The code used to conduct the different analyses is available in the
supplementary data and at the following GitHub repository: [https://github.com/Bertrand-](https://github.com/Bertrand-Ornella/Brain-evolution-Sciuroidea)
[Ornella/Brain-evolution-Sciuroidea](https://github.com/Bertrand-Ornella/Brain-evolution-Sciuroidea).

**Author contributions**

OCB conceived and designed the study. OCB and MTS acquired the dataset. OCB and
SLB wrote the manuscript. Analyses were performed by OCB, HPP and JAS. All
authors carried out interpretations, revised the manuscript and provided final approval
before submission.

**Competing interests.** We have no competing interests.

**Acknowledgments**

We thank Dr. Hans Larsson from McGill University in Montreal (Quebec) for his
generous help and for facilitating the scanning process of the three new squirrel
specimens, and Rui Tahara for scanning the specimens at the Integrative Quantitative
Biology Initiative (McGill University, QB). We also thank the American Museum of
Natural History (AMNH) Mammalogy collection team in New York for sending the
specimens to Montreal for them to be scanned. This research was funded by a
Marie Skłodowska-Curie Actions: Individual Fellowship (H2020-MSCA-IF-2018-2020;
No. 792611) to OCB; a European Research Council (ERC) Starting Grant (No. 756226)
to SLB, under the European Union's Horizon 2020 Research and Innovation
Programme; the National Agency for Research and Development (ANID)/
PFCHA/Doctorado en el extranjero Becas Chile/2018-72190003 to HPP; a Leverhulme
Trust Research Project grant (RPG-2017-167) to PI S.L.B., which funds JAS; an
NSERC Discovery Grant to MTS. We would like to thank Phil Cox and two anonymous
reviewers for their valuable feedback that improved this manuscript.

[revised manuscript text omitted]

966 104 Shultz, S. & Finlayson, L. V. Large body and small brain and group sizes are associated with 967 predator preferences for mammalian prey. *Behav. Ecol.* **21**, 1073-1079, 968 doi:10.1093/beheco/arq108 (2010).

969 105 Catania, K. C., Lyon, D. C., Mock, O. B. & Kaas, J. H. Cortical organization in shrews: evidence 970 from five species. *J. Comp. Neurol.* **410**, 55-72 (1999).

971 106 Catania, K. C. & Kaas, J. H. Organization of the somatosensory cortex of the star-nosed mole. *J. 972 Comp. Neurol.* **351**, 549-567, doi:10.1002/cne.903510406 (1995).

Stephan, H., Baron, G. & Frahm, H. D. *Insectivora: with a stereotaxic atlas of the hedgehog brain*.
Vol. 1 573 p. (Springer Science & Business Media, 2012).

Williams, R. W. & Herrup, K. The control of neuron number. *Annu. Rev. Neurosci.* **11**, 423-453
(1988).

Krubitzer, L., Campi, K. L. & Cooke, D. F. All rodents are not the same: a modern synthesis of
cortical organization. *Brain Behav. Evol.* **78**, 51-93, doi:10.1159/000327320 (2011).

Michener, G. in *Advances in the study of mammalian behavior* Vol. 7 528-572 (1983).

Matějů, J. & Kratochvíl, L. Sexual size dimorphism in ground squirrels (Rodentia: Sciuridae:
Marmotini) does not correlate with body size and sociality. *Frontiers in Zoology* **10**, 27,
doi:10.1186/1742-9994-10-27 (2013).

Falk, D. in *Handbook of Paleoanthropology* (eds Winfried Henke & Ian Tattersall) 1495-1525
(Springer Berlin Heidelberg, 2015).

Voogd, J. & Wylie, D. R. Functional and anatomical organization of floccular zones: a preserved
feature in vertebrates. *J. Comp. Neurol.* **470**, 107-112 (2004).

Waespe, W., Cohen, B. & Raphan, T. Role of the flocculus and paraflocculus in optokinetic
nystagmus and visual-vestibular interactions: effects of lesions. *Exp. Brain Res.* **50**, 9-33 (1983).

Lee, C. & Goh, H. Rodent species in mangosteen intercropped with coconuts and their bait
preferences. *Journal of tropical agriculture and food science* **28**, 31-38 (2000).

O'Shea, T. J. Home Range, Social Behavior, and Dominance Relationships in the African
Unstriped Ground Squirrel, *Xerus rutilus*. *J. Mammal.* **57**, 450-460, doi:10.2307/1379295 (1976).

Yasuda, M., Miura, S. & Hussein, N. A. Evidence for food hoarding behaviour in terrestrial
rodents in Pasoh Forest Reserve, a Malaysian lowland rain forest. *Journal of Tropical Forest
Science* **12**, 164-173 (2000).

Noda, N. & Mikami, A. Discharges of neurons in the dorsal paraflocculus of monkeys during eye
movements and visual stimulation. *J. Neurophysiol.* **56**, 1129-1146,
doi:10.1152/jn.1986.56.4.1129 (1986).

Kralj-Hans, I., Baizer, J. S., Swales, C. & Glickstein, M. Independent roles for the dorsal
paraflocculus and vermal lobule VII of the cerebellum in visuomotor coordination. *Exp. Brain
Res.* **177**, 209-222 (2007).

Bishop, K. L. The relationship between 3-D kinematics and gliding performance in the southern
flying squirrel, *Glaucomys volans*. *J. Exp. Biol.* **209**, 689-701, doi:10.1242/jeb.02062
(2006).

Hayssen, V. Patterns of Body and Tail Length and Body Mass in Sciuridae. *J. Mammal.* **89**, 852-
873, doi:10.1644/07-mamm-a-217.1 (2008).

Lu, X., Ge, D., Xia, L., Huang, C. & Yang, Q. Geometric morphometric study of the skull shape
diversification in Sciuridae (Mammalia, Rodentia). *Integr Zool* **9**, 231-245, doi:10.1111/1749-
4877.12035 (2014).

Gunnell, G. F., Rose, K. D. & Rasmussen, D. T. in *Evolution of Tertiary Mammals of North
America: Volume 2: Small Mammals, Xenarthrans, and Marine Mammals* 239-262 (Cambridge
University Press, 2008).

Zanazzi, A., Kohn, M. J., MacFadden, B. J. & Terry, D. O. Large temperature drop across the
Eocene–Oligocene transition in central North America. *Nature* **445**, 639-642,
doi:10.1038/nature05551 (2007).

Shattuck, M. R. & Williams, S. A. Arboreality has allowed for the evolution of increased longevity
in mammals. *Proceedings of the National Academy of Sciences* **107**, 4635-4639,
doi:10.1073/pnas.0911439107 (2010).

Dunbar, R. I. M. The social brain hypothesis. *Evolutionary Anthropology: Issues, News, and
Reviews: Issues, News, and Reviews* **6**, 178-190 (1998).

Boyer, D. M., Gunnell, G. F., Kaufman, S. & McGeary, T. M. MORPHOSOURCE: ARCHIVING AND
SHARING 3-D DIGITAL SPECIMEN DATA. *The Paleontological Society Papers* **22**, 157-181,
doi:10.1017/scs.2017.13 (2016).

**Figures**

**Figure 1.** Virtual endocast of the early Oligocene squirrel *Cedromus wilsoni* (USNM
256584) based on computed tomography (CT) data **a.** lateral view inside a solid
cranium; **b.** lateral view inside a translucent cranium; **c.** lateral view with the different
regions estimated in the paper highlighted; **d.** dorsal view; **e.** ventral view.

**Abbreviations:** **nx**, neocortex; **ob**, olfactory bulb; **pl**, petrosal lobule. Scale bars equal
10 mm.

**Figure 2.** Boxplots of the **a.** Phylogenetic Encephalization Quotient (PEQ); **b.**
 Neocortical surface area percentage; **c.** petrosal lobule volume percentage; **d.** olfactory
 bulb volume percentage for our sample of rodents categorized by locomotor mode. * are
 for categories that are significantly different based on the Asymptotic K-Sample Fisher-
 Pitman Permutation Test (Table 1).

**Figure 3.** Phylogenetic corrected PGLS regressions of the **a.** Endocranial volume (cm^3)
vs. Body mass (g) ; **b.** Neocortical surface area (mm^2) vs. Endocranial surface area
(mm^2) **c.** Olfactory bulbs (mm^3) vs. Body mass (mg); **d.** Olfactory bulbs (mm^3) vs.
Endocranial volume (mm^3); Petrosal lobule volume (mm^3) vs. Body mass (mg); **e.**
Petrosal lobule volume (mm^3) vs. Endocranial volume (mm^3). Data used for all
regression are in Logarithm 10. **Abbreviations:** 1.EB, Endocranial volume ~ Body mass
+ Locomotion * Body mass; 1.PB, Petrosal lobule volume ~ Body mass +
Locomotion*Body mass; 1.PE, Petrosal lobule volume ~ Endocranial volume +
Locomotion; 2.NE, Neocortex surface area ~ Endocranial surface area + Locomotion;
2.OB, Olfactory bulb volume ~ Body mass + Locomotion; 2.OE, Olfactory bulb volume ~
Endocranial volume + Locomotion.

**Figure 4.** Ancestral state reconstruction of the Phylogenetic Encephalization Quotient

(PEQ) with a geological time scale. The temperature and oxygen level ($\delta^{18}\text{O}$) plot

during the Cenozoic is modified from Zachos et al.⁴⁰.

**Figure 5.** Ancestral state reconstructions with a geological timescale. **a.** Neocortical
 surface area percentage; **b.** Petrosal lobule volume percentage.

**Figure 6.** Ancestral state reconstruction of the locomotor behaviour of the taxa used in

the current study with a geological timescale.

**Table 1.** Asymptotic K-Sample Fisher-Pitman Permutation Test and pairwise
 comparisons performed on the Phylogenetic Quotient (PEQ), the olfactory bulb volume,
 petrosal lobule volume and neocortical surface area percentages. **Abbreviation:** df,
 degrees of freedom.

	PEQ			Petrosal lobules		
Permutation test	$\chi^2 = 17.166$, df = 4, p-value = 0.001794			$\chi^2 = 17.39$, df = 4, p-value = 0.001623		
Pairwise comparisons	Stat	p.value	p.adjust	Stat	p.value	p.adjust
Arboreal - Fossorial	3.143	0.002	0.017	3.163	0.002	0.016
Arboreal - Glider	0.746	0.456	0.506	2.603	0.009	0.042
Arboreal - Scansorial	2.319	0.020	0.051	1.873	0.061	0.100
Arboreal - Terrestrial	2.065	0.039	0.078	1.935	0.053	0.100
Fossorial - Glider	-2.721	0.007	0.033	-1.812	0.070	0.100
Fossorial - Scansorial	-2.414	0.016	0.051	-2.498	0.012	0.042
Fossorial - Terrestrial	-1.987	0.047	0.078	-1.953	0.051	0.100
Glider - Scansorial	1.650	0.099	0.141	-1.474	0.141	0.176
Glider - Terrestrial	1.459	0.145	0.181	-0.836	0.403	0.448
Scansorial - Terrestrial	0.106	0.916	0.916	0.457	0.647	0.647

	Neocortex			Olfactory bulbs		
Permutation test	$\chi^2 = 19.879$, df = 4, p-value = 0.0005277			$\chi^2 = 11.424$, df = 4, p-value = 0.02219		
Pairwise comparisons	Stat	p.value	p.adjust	Stat	p.value	p.adjust
Arboreal - Fossorial	3.569	0.000	0.003	-2.416	0.016	0.112
Arboreal - Glider	-0.519	0.604	0.671	0.479	0.632	0.632
Arboreal - Scansorial	2.045	0.041	0.101	-2.014	0.044	0.112
Arboreal - Terrestrial	0.081	0.935	0.935	-1.296	0.195	0.278
Fossorial - Glider	-3.441	0.001	0.003	2.284	0.022	0.112
Fossorial - Scansorial	-1.217	0.223	0.319	0.773	0.439	0.488
Fossorial - Terrestrial	-3.242	0.001	0.004	1.403	0.161	0.268
Glider - Scansorial	1.956	0.050	0.101	-2.007	0.045	0.112
Glider - Terrestrial	1.014	0.311	0.389	-1.490	0.136	0.268
Scansorial - Terrestrial	-1.678	0.093	0.156	0.822	0.411	0.488

**Table 2.** Regressions with selected model and formula. **Abbreviation:** Pred., predictor variable.

Regression	Model	Formula	Pred. 1	Pred. 2	Pred.1*Pred. 2
1.EB	PGLS - Lambda	Endocranial volume ~ Body mass + Locomotion * Body mass	Body mass	Locomotion	Body mass*Locomotion
1.PB	PGLS - Lambda	Petrosal lobule volume ~ Body mass + Locomotion*Body mass	Body mass	Locomotion	Body mass*Locomotion
1.PE	PGLS - Lambda	Petrosal lobule volume ~ Endocranial volume + Locomotion	Endocranial volume	Locomotion	-
2.NE	PGLS - Brownian	Neocortex surface area ~ Endocranial surface area + Locomotion	Endocranial volume	Locomotion	-
2.OB	PGLS - OU	Olfactory bulb volume ~ Body mass + Locomotion	Body mass	Locomotion	-
2.OE	PGLS - Lambda	Olfactory bulb volume ~ Endocranial volume + Locomotion	Endocranial volume	Locomotion	-

**Table 3.** Results from the different PLGS regression analyses. **Abbreviations:** 1.EB, Endocranial volume ~ Body mass +
 Locomotion * Body mass; 1.PB, Petrosal lobule volume ~ Body mass + Locomotion*Body mass; 1.PE, Petrosal lobule
 volume ~ Endocranial volume + Locomotion; 2.NE, Neocortex surface area ~ Endocranial surface area + Locomotion;
 2.OB, Olfactory bulb volume ~ Body mass + Locomotion; 2.OE, Olfactory bulb volume ~ Endocranial volume +
 Locomotion; RSE, Residual standard error; Df, degrees of freedom.

Regression	Intercept	slope (Pred. 1)	slope (Pred. 2)	slope (Pred.1*Pred. 2)	P-value (Pred. 1)	P-value (Pred. 2)	P-value (Pred. 1*Pred. 2)	Lambda	RSE	Df
1.EB	-1.075	0.691	0.077	-0.042	0.000	0.121	0.024	0.722	0.103	38, 34
1.PB	-1.986	0.705	0.446	-0.091	0.000	0.056	0.027	0.535	0.189	37, 33
1.PE	-1.115	0.825	-0.046	-	0.000	0.009	-	0.544	0.171	37, 34
2.NE	-0.503	0.976	-0.014	-	0.000	0.001	-	1.008	0.130	38, 35
2.OB	-0.625	0.511	-0.014	-	0.000	0.332	-	0.293	0.122	38, 35
2.OE	-0.927	0.862	0.017	-	0.000	0.182	-	0.690	0.149	38, 35

**Table 4.** Non-phylogenetically corrected post-hoc pairwise comparisons P-values for
 the different locomotor modes. **Abbreviations:** 1.EB, Endocranial volume ~ Body mass
 + Locomotion * Body mass; 1.PB, Petrosal lobule volume ~ Body mass +
 Locomotion*Body mass; 1.PE, Petrosal lobule volume ~ Endocranial volume +
 Locomotion; 2.NE, Neocortex surface area ~ Endocranial surface area + Locomotion;
 2.OB, Olfactory bulb volume ~ Body mass + Locomotion; 2.OE, Olfactory bulb volume ~
 Endocranial volume + Locomotion; OB, olfactory bulbs; PL, petrosal lobules.

Regression 1.EB	Arboreal	Glider	Scansorial	Terrestrial	Summary
Glider	0.523				Arboreal & Glider: Relatively larger brain
Scansorial	0.001	0.003			
Terrestrial	0.009	0.020	0.803		
Fossorial	0.001	0.011	0.806	0.980	
Regression 1.PB					
Glider	0.239				Large gliders: absolutely larger PL compared to small gliders
Scansorial	0.166	0.004			
Terrestrial	0.026	0.002	0.260		
Fossorial	0.064	0.001	0.518	0.606	
Regression 1.PE					
Glider	0.008				Fossorial and Glider: Relatively smaller PL. Arboreal & Scansorial: Relatively larger PL
Scansorial	0.131	0.224			
Terrestrial	0.086	0.420	0.751		
Fossorial	0.001	0.401	0.027	0.068	
Regression 2.NE					
Glider	0.749				Fossorial & Scansorial: Relatively smaller neocortex
Scansorial	0.005	0.019			
Terrestrial	0.967	0.766	0.010		
Fossorial	0.003	0.022	0.635	0.003	
Regression 2.OE					
Glider	0.595				Arboreal & Glider: Relatively smaller OB
Scansorial	0.063	0.020			
Terrestrial	0.280	0.160	0.542		
Fossorial	0.007	0.010	0.445	0.179	

1096

Reviewers' comments:

Reviewer #1 (Remarks to the Author):

The authors have clearly spent a lot of time revising this manuscript and it is much improved as a result. The major concerns that we had in our original review have now been addressed i.e. the statistical analyses. The use of permutation tests and PGLS models has much greater validity, I think, than the analyses in the original manuscripts, and thus the conclusions that are drawn are much more robust. The more sophisticated models used here also address the concerns that we originally had about teasing apart the influences of body mass and locomotion on the brain.

I just have a few more minor issues that need clearing up, listed below. Otherwise I am satisfied that the authors have addressed all my concerns.

Thank you so much for your feedback. We have addressed the additional issues.

Line 139: Van Valkenburgh misspelt (final H missing)

Line 139: The change was made.

Line 230: From Table 1, it seems you have used some kind of multiple comparisons adjustment. This needs to be described here (which method of correction did you use?).

Lines 247-249: We added a description of the correction that we used. We implemented the “BH” also known as “fdr” correction that controls for false discovery rate.

Benjamini Y, Hochberg Y (1995) Controlling the False Discovery Rate: A Practical and Powerful Approach to Multiple Testing. *Journal of the Royal Statistical Society: Series B (Methodological)* 57 (1):289-300. doi:<https://doi.org/10.1111/j.2517-6161.1995.tb02031.x>

Line 259: Would be good to list all the equations tested in a supplementary table alongside their AIC values.

Lines 290-291: We added the AIC values for each model that was selected in **Table 2**. All the AIC values for each equation are available in the code section of the supplementary data so we decided to not make an additional supplementary table to avoid redundancy. We clarified in the text where the values can be found.

Table 2. Results from the PGLS regression analyses. **Abbreviations:** 1.EB, Endocranial volume ~ Body mass + Locomotion * Body mass; 1.PB, Petrosal lobule volume ~ Body mass + Locomotion*Body mass; 2.PE, Petrosal lobule volume ~ Endocranial volume + Locomotion; 2.NE, Neocortex surface area ~ Endocranial surface area + Locomotion; 1.OB, Olfactory bulb volume ~ Body mass + Locomotion; 2.OE,

Olfactory bulb volume ~ Endocranial volume + Locomotion; **AIC, Akaike Information Criterion**; Df, degrees of freedom; RSE, Residual standard error.

Regression	Model - PGLS	AIC	Intercept	slope (Pred. 1)	slope (Pred. 2)	slope (Pred.1* Pred. 2)	P-value (Pred. 1)	P-value (Pred. 2)	P-value (Pred. 1*Pred. 2)	Lambda	RSE	Df
1.EB	Lambda	-79.080	-1.075	0.691	0.077	-0.042	0.000	0.121	0.024	0.722	0.103	38, 34
1.PB	Lambda	-20.372	-1.986	0.705	0.446	-0.091	0.000	0.056	0.027	0.535	0.189	37, 33
2.PE	Lambda	-30.028	-1.115	0.825	-0.046	-	0.000	0.009	-	0.544	0.171	37, 34
2.NE	Brownian	-120.954	-0.503	0.976	-0.014	-	0.000	0.001	-	1.008	0.130	38, 35
1.OB	Ornstein-Uh	-47.930	-0.625	0.511	-0.014	-	0.000	0.332	-	0.293	0.122	38, 35
2.OE	Lambda	-50.479	-0.927	0.862	0.017	-	0.000	0.182	-	0.690	0.149	38, 35

Lines 277-289: I think you've made an error in the abbreviations here – shouldn't the final one be 2OE (i.e. set 2, Olfactory bulb, Endocranial volume)? See also comment on Tables 2-3 below.

Line 300: Yes, this is correct. We made an error, thank you. We have made the change (see **Table 2** above). Tables 2 and 3 were combined following the suggestion from Reviewer 2.

Line 558: 'lead' should be 'led' or 'leads'

Line 582: Thank you, we made this correction.

Line 564: There seems to be a word missing after 'terrestrial' – the other two terrestrial what?

Line 588: We added the word "rodents".

Line 600: Probably worth noting here that many mammalian species lack the petrosal lobule / subarcuate fossa (including some other rodents and ourselves), but presumably still retain the functional elements of those structures since they are able to stabilize their gaze. See Gannon et al (1988, Am J Phys Anthropol, 77: 143-164) for a list of species lacking the subarcuate fossa.

Lines 625-629: Thank you for the suggestion. We added a sentence clarifying our thoughts.

The data that Gannon et al. (1988) mentioned were gathered by Flower (1870). They didn't find subarcuate fossa in elephants, caviomorph rodents, rhinos and apes.

Overall, Gannon et al. (1988) were not testing whether the lack of petrosal lobules in species in which they are missing was related to locomotor behaviour or ecology in general. So, we are reluctant at the idea of adding a sentence about inconclusive results that contradict our findings.

Regarding other rodents, we found more recent papers on caviomorphs showing that the fossorial caviomorph *Ctenomys* has smaller petrosal lobules compared to semi-aquatic caviomorphs (Arnaudo et al., 2020).

Flower WH (1870) An Introduction to the Osteology of the Mammalia. London: Macmillan and Co.

Gannon PJ, Eden AR, Laitman JT (1988) The subarcuate fossa and cerebellum of extant primates: Comparative study of a skull-brain interface. *Am J Phys Anthropol* 77 (2):143-164.
doi:<https://doi.org/10.1002/ajpa.1330770202>

Arnaudo ME, Arnal M, Ekdale EG (2020) The auditory region of a caviomorph rodent (Hystricognathi) from the early Miocene of Patagonia (South America) and evolutionary considerations. *J Vert Paleontol* 40 (2):e1777557. doi:10.1080/02724634.2020.1777557

Tables 2-3: I think there's a mistake in the abbreviations (although a different one from above). Shouldn't 2OB, be 1OB? All the equations that use body mass as a predictor should be set 1, right? That would make the tables less confusing because the regressions would be presented in a more logical order i.e. 1EB, 1PB, 1OB, 2NE, 2PE, 2OE.

Tables 2-3, S6: Yes, this is correct. We made an error, thank you. We have made the change.

Philip Cox
University of York

Reviewer #2 (Remarks to the Author):

Thank you again for the opportunity to review the revised manuscript entitled "The impact of locomotion on the brain evolution of squirrels and close relatives". I appreciate the authors addressing all of my comments. The work they have done on the manuscript is commendable and their clarifications regarding my concerns appear logical and reasonable. The manuscript is well-researched and the study is supported with a strong rationale. I have just a few additional points to bring up.

Thank you so much for your feedback. We have addressed the additional issues (see below).

MAIN POINTS

The authors have done a great job revising their tests. And for the most part, these are well done. However, there are some choices that could perhaps benefit with some further consideration. I have listed these below with some other brief notes.

Abstract:

Suggest including (1) how many species were used, and (2) the use of digital

endocasts.

The word count can be reduced by removing word clusters like “and three components”, “we find that”, etc.

Line 16: We added the suggestions and deleted “and three components”.

Introduction:

This is very much improved.

Methods:

The permutational tests make little sense to me from the descriptions from line 230.

There are no other studies cited in this section that provide grounds for using them, so I’m not following their purpose.

Why are these tests better than running the same regressions as for later tests, but on Locomotion as a single predictor?

A concern regarding Fisher-Pitman Permutation tests is that in order to compare group means, this test requires homogeneity of variances across groups (Boik, 1987). This is because Fisher-Pitman Permutation tests can identify any types of differences between group distributions and so it may not necessarily demonstrate a difference between means if there is a significant difference between the variances (Neuhäuser & Manly, 2004).

Some of the distributions in Figure 2 suggest that variances between some groups are not homogeneous. But I suppose variances can be compared with an F-test or something similar to verify the conditions.

Boik, R. J. (1987). The Fisher-Pitman permutation test: A non-robust alternative to the normal theory F test when variances are heterogeneous. *British Journal of Mathematical and Statistical Psychology*, 40(1), 26–42.

Neuhäuser, M., & Manly, B. F. (2004). The Fisher-Pitman permutation test when testing for differences in mean and variance. *Psychological reports*, 94(1), 189-194.

Lines 231-243 and line 363: We decided to use permutation tests as Reviewer 1 suggested because our data might not be normally distributed. Also, when the sample size of a group is small, permutation tests are recommended over *t* and *F* tests (Ludbrook and Dudley, 1998).

However, we understand the concern of Reviewer 2 and we checked for the homogeneity of the variances for the PEQ, olfactory bulb volume, petrosal lobule

volume, and neocortical surface area percentages. We found that homogeneity of the variances was verified for all except for the neocortical surface percentage.

The code to determine the normality of the data and the homogeneity of the variances has been added in the code section of the supplementary data. We also added in the text that we performed these tests and what resulted from them. We acknowledged that the results from the neocortex should be interpreted with caution in case group means are significantly different for the neocortical surface area. The boxplot for the neocortical surface area shows that the group means that are significantly different are also visually quite different with absolutely not overlap between groups. Therefore, we think that this violation to the assumption should not affect our results about the neocortex.

Regardless, we are now completely up front about these tests, why we used them, and their potential limitation. We also note that they are one of many statistical tests we use in this manuscript and our main conclusions do not hinge on them alone.

Ludbrook J, Dudley H (1998) Why Permutation Tests Are Superior to t and F Tests in Biomedical Research. *The American Statistician* 52 (2):127-132. doi:10.2307/2685470

In the paragraph of lines 254-265, the authors state that they choose not to incorporate phylogenetic relations to test all models; their reason being that “in some instance OLS outperforms phylogenetic generalized least squares (PGLS)”. They assessed whether a PGLS analysis is appropriate for each model by plotting the residuals, followed by what seems to be a visual assessment of the residual distributions. While an OLS may perform better sometimes, I would still argue that all tests should be consistent for each model.

Firstly, they are different tests, so the results for each aren't strictly comparable. Also, a subjective decision based on visual inspection may miss something that is not immediately apparent in the eye-balling process - even mild trends that are hard to see can sometimes be statistically relevant.

Finally, if no phylogenetical signal is evident, then relatedness won't significantly impact the outputs anyway. I think it would be more thorough and reliable to either run both OLS and PGLS or just PGLS alone on all models, rather than choosing one or the other.

Lines 278-279: We understand the concern of Reviewer 2 and agree that OLS and PGLS should not be directly compared. Our revision reflected this. This is why in this new version, we ran and kept the results from both OLS and PGLS separated, meaning that we did not make direct comparisons between the results of the PGLS and OLS.

We first ran OLS on all regressions and found that there was a phylogenetic signal in all cases. So, we re-ran all of our analyses using PGLS and in all cases PGLS was selected for the rest of the analyses. **Figure S2** illustrates our methodological steps that were applied first for OLS and when we deduced that a phylogenetic signal existed, we follow the same steps and re-ran all analyses using PGLS.

The authors cite Weisbecker et al. for not performing phylogenetic pairwise comparisons (line 302). Which is fine. But if the authors are really interested in pairwise comparisons on phylogenetically adjusted distributions, this may be possible using the 'phylANOVA' function in the phytools package. This function performs a simulation-based phylogenetic ANOVA and can then carry out posthoc comparisons of means among groups. However, this function only permits a single predictor variable containing groups. It could be used with Locomotion alone, but with additional variables is tricky. Body size could possibly be included as an initial predictor by using the residuals of the body size/measurement (e.g. petrosal lobules) correlation as the response variable and Locomotion as the predictor. Just some food for thought.

Lines 323-332: Thank you for suggesting this method. We indeed considered using the phylANOVA function from the phytools package but as Reviewer 2 explains this function only allows one predictor variable and we wanted to be able to apply the phylogenetic correction to all variables in our multivariate study, as we are looking at more than just locomotion.

The idea of using the residuals of the body mass/measurement would work for 1.OB (Olfactory bulb volume ~ Body mass + Locomotion) but would not work for the 1.EB (Endocranial volume ~ Body mass + Locomotion * Body mass) and 1.PB (Petrosal lobule volume ~ Body mass + Locomotion*Body mass) because both include an extra predictor: the interaction between body mass and locomotion. This interaction could not be incorporated into the residuals.

We added a few sentences explaining our choice to use non-phylogenetic pairwise comparison tests.

There are also some sections of the methods that could be streamlined. This is more an editing issue, so I won't list things at length. An example, however, would be combining lines 239-243 and lines 243-247. These are both lengthy sentences that essentially provide the same information.

Lines 252-260: We agree with Reviewer 2 that these sentences essentially provide the same information, but we wanted to make sure that our methodology was clearly explained. We think it would be hard to combine both sentences without taking some crucial information away. If the editor prefers that we combine both sentences, we will make the change.

Results

These are fine, based on the tests performed.

Discussion/Conclusion:

These are also nice. The conclusion rounds everything up very well.

MINOR COMMENTS

Line 229: “Additionally, we also obtained the residuals from the PGLS regression endocranial volume vs body mass.”

--This feels like half a sentence to me. Why was this mentioned? What was done with the residuals once they were obtained?

Line 228: Nothing was done with the residuals because they are essentially the same as the PEQ values. We understand the reviewer’s concern and we decided to delete this sentence.

Line 232: “Phylogeny was only considered here in the calculation of the PEQ.”

--Should mention why/why not for the others

Line 244: We deleted this sentence because we feel that it might lead to unnecessary confusion. We just had this sentence to say that PEQ is the phylogenetic Encephalization Quotient so includes phylogeny in it. The other boxplots don’t because they represent the percentage of different regions of the brain.

Lines 279-286: The regressions in the methods are listed as 1EB, 1PB, 1OB, 2NE, 2PE, 2OB.

But in the Tables as 1EB, 1PB, 1PE, 2NE, 2OB, 2OE. I think these models do not correspond. Some are olfactory bulbs while others are petrosal lobules.

Lines 294-301: We made the necessary corrections, so the test and tables are now the same in terms of the abbreviations.

Line 349: “p-value= 0.000”

--Remove ‘-value’ here and elsewhere throughout the results (except table captions for defining “p”), and instead write as ‘p < 0.001’ in this instance.

Line 374: We removed ‘-value’ everywhere in the text. We change the p-value in Table 1 to p < 0.001 for the Neocortex, p < 0.002 for the PEQ and petrosal lobules and p < 0.02 for the olfactory bulbs.

Table 1. Asymptotic K-Sample Fisher-Pitman Permutation Test and pairwise comparisons performed on the Phylogenetic Quotient (PEQ), the olfactory bulb volume, petrosal lobule volume and neocortical surface area percentages. **Abbreviation:** df, degrees of freedom.

	PEQ			Petrosal lobules		
Permutation test	$\chi^2 = 17.166$, df = 4, p < 0.002			$\chi^2 = 17.39$, df = 4, p < 0.002		
Pairwise comparisons	Stat	p.value	p.adjust	Stat	p.value	p.adjust
Arboreal - Fossorial	3.143	0.002	0.017	3.163	0.002	0.016
Arboreal - Glider	0.746	0.456	0.506	2.603	0.009	0.042
Arboreal - Scansorial	2.319	0.020	0.051	1.873	0.061	0.100
Arboreal - Terrestrial	2.065	0.039	0.078	1.935	0.053	0.100
Fossorial - Glider	-2.721	0.007	0.033	-1.812	0.070	0.100
Fossorial - Scansorial	-2.414	0.016	0.051	-2.498	0.012	0.042
Fossorial - Terrestrial	-1.987	0.047	0.078	-1.953	0.051	0.100
Glider - Scansorial	1.650	0.099	0.141	-1.474	0.141	0.176
Glider - Terrestrial	1.459	0.145	0.181	-0.836	0.403	0.448
Scansorial - Terrestrial	0.106	0.916	0.916	0.457	0.647	0.647
	Neocortex			Olfactory bulbs		
Permutation test	$\chi^2 = 19.879$, df = 4, p < 0.001			$\chi^2 = 11.424$, df = 4, p < 0.02		
Pairwise comparisons	Stat	p.value	p.adjust	Stat	p.value	p.adjust
Arboreal - Fossorial	3.569	0.000	0.003	-2.416	0.016	0.112
Arboreal - Glider	-0.519	0.604	0.671	0.479	0.632	0.632
Arboreal - Scansorial	2.045	0.041	0.101	-2.014	0.044	0.112
Arboreal - Terrestrial	0.081	0.935	0.935	-1.296	0.195	0.278
Fossorial - Glider	-3.441	0.001	0.003	2.284	0.022	0.112
Fossorial - Scansorial	-1.217	0.223	0.319	0.773	0.439	0.488
Fossorial - Terrestrial	-3.242	0.001	0.004	1.403	0.161	0.268
Glider - Scansorial	1.956	0.050	0.101	-2.007	0.045	0.112
Glider - Terrestrial	1.014	0.311	0.389	-1.490	0.136	0.268
Scansorial - Terrestrial	-1.678	0.093	0.156	0.822	0.411	0.488

Line 556-559: “Knowing the role of the petrosal lobules in maintaining the position of the eyes when the head rotates (vestibulo-ocular reflex) and in the tracking of moving object (smooth pursuit) lead us to hypothesize that these functions may be crucial when living in the 3D complex environment of the trees.

--This sentence could be more impactful in the Introduction. Perhaps somewhere in paragraph 80-88, which would lead to the question regarding their relative sizes across species/Locomotion groups.

Lines 580-583: We prefer to keep this sentence in the discussion as it lays the foundations for the rest of the paragraph that follows. The paragraph in the introduction (lines 80-88) is intended to briefly review what has been done and going into specifics about the petrosal lobules would lead to more details required for other parts of the brain (like the neocortex, for instance), which would make the introduction considerably longer and less streamlined. We prefer to keep this section short, but if the editor would like us to move the sentence, we will make the change.

Line 1029: Figure 1 is still missing an ‘e’ in the image for part e.

I would also encourage referring to this figure in the introduction so that the reader can see what an endocast is and get their bearings on the different parts.

Lines 40 and 1072: We have added a mention of **Figure 1** in the introduction and we have added “e” in Figure 1.

Figure 1. Virtual endocast of the early Oligocene squirrel *Cedromus wilsoni* (USNM 256584) based on computed tomography (CT) data **a.** lateral view inside a solid cranium; **b.** lateral view inside a translucent cranium; **c.** lateral view with the different regions estimated in the paper highlighted; **d.** dorsal view; **e.** ventral view. **Abbreviations:** **nx**, neocortex; **ob**, olfactory bulb; **pl**, petrosal lobule. Scale bars equal 10 mm.

Line 1042: I think there is a lot of repeated information here between the figure and the caption. Perhaps replace the section definitions with the abbreviations since the definitions are also the axis labels.

Line 1082: We simplified our legend of **Figure 3** and changed the definition with the abbreviations as suggested by Reviewer 2.

Figure 3. Phylogenetic corrected PGLS regressions of **a.** 1.EB, Endocranial volume ~ Body mass + Locomotion * Body mass; **b.** 2.NE, Neocortex surface area ~ Endocranial surface area + Locomotion; **c.** 1.PB, Petrosal lobule volume ~ Body mass + Locomotion * Body mass; **d.** 2.PE, Petrosal lobule volume ~ Endocranial volume + Locomotion; **e.** 1.OB, Olfactory bulb volume ~ Body mass + Locomotion; **f.** 2.OE, Olfactory bulb volume ~ Endocranial volume + Locomotion. Data used for all regression are in Logarithm 10.

Line 1076: I'm not sure this table is necessary. The "model" column can be incorporated into Table 3 and the rest is detailed in the caption for Table 3 anyway.

Line 1110: We merged Tables 2 and 3. We deleted Table 2 and incorporated the "model" column to Table 3 (now Table 2).

Table 2. Results from the PGLS regression analyses. **Abbreviations:** 1.EB, Endocranial volume ~ Body mass + Locomotion * Body mass; 1.PB, Petrosal lobule volume ~ Body mass + Locomotion*Body mass; 2.PE, Petrosal lobule volume ~ Endocranial volume + Locomotion; 2.NE, Neocortex surface area ~ Endocranial surface

area + Locomotion; **1.OB**, Olfactory bulb volume ~ Body mass + Locomotion; **2.OE**, Olfactory bulb volume ~ Endocranial volume + Locomotion; **AIC**, Akaike Information Criterion; Df, degrees of freedom; RSE, Residual standard error.

Regression	Model - PGLS	AIC	Intercept	slope (Pred. 1)	slope (Pred. 2)	slope (Pred.1* Pred. 2)	P-value (Pred. 1)	P-value (Pred. 2)	P-value (Pred. 1*Pred. 2)	Lambda	RSE	Df
1.EB	Lambda	-79.080	-1.075	0.691	0.077	-0.042	0.000	0.121	0.024	0.722	0.103	38, 34
1.PB	Lambda	-20.372	-1.986	0.705	0.446	-0.091	0.000	0.056	0.027	0.535	0.189	37, 33
2.PE	Lambda	-30.028	-1.115	0.825	-0.046	-	0.000	0.009	-	0.544	0.171	37, 34
2.NE	Brownian	-120.954	-0.503	0.976	-0.014	-	0.000	0.001	-	1.008	0.130	38, 35
1.OB	Ornstein-Uh	-47.930	-0.625	0.511	-0.014	-	0.000	0.332	-	0.293	0.122	38, 35
2.OE	Lambda	-50.479	-0.927	0.862	0.017	-	0.000	0.182	-	0.690	0.149	38, 35

Line 1079: PGLS, not PLGS. And suggest rewording the title. What's "different" about this analysis?

Line 1110: We corrected 'PGLS' and deleted 'different' in the legend for Table 2 (see above).

Reviewer #3 (Remarks to the Author):

Please forgive me for my delay in returning this review, especially given how short my re-review has turned out to be. 2021 continues to be as weird as 2020.

I reviewed a previous iteration of this manuscript. My primary concerns related to the lack of clarity regarding the statistical analyses and whether the analyses the authors used were, in fact, most appropriate. The manuscript is very much improved in clarity and I believe that it should be accepted for publication. It is an excellent and interesting work.

The questions are now clearly laid out in the introduction. This greatly enhanced my understanding of the goals of the study.

The authors have applied a new series of appropriate models for the questions posed in the manuscript and the methods are quite clear. It appears to address all of my original concerns.

I have made a very small number of specific comments on the PDF. Take them or leave them.

Thank you for the excellent read.

Response to Reviewer 3 comments from the PDF:

Line 84: groups or among these animals

Line 84: We changed 'across' to 'among'.

Line 479: what is meant here? Referring to 'genealogy'

Line 503: We deleted 'genealogy'.

Line 509: citation?

Lines 530-533: We don't think that a citation is necessary here.